# WORLD-IN-WORLD: WORLD MODELS IN A CLOSED-LOOP WORLD

**Jiahan Zhang**[1,*] **Muqing Jiang**[2,*] **Nanru Dai**[1] **Taiming Lu**[1,3] **Arda Uzunoglu**[1] **Shunchi Zhang**[1]
**Yana Wei**[1] **Jiahao Wang**[1] **Vishal M. Patel**[1] **Paul Pu Liang**[4] **Daniel Khashabi**[1] **Cheng Peng**[1]
**Rama Chellappa**[1] **Tianmin Shu**[1] **Alan Yuille**[1] **Yilun Du**[5] **Jieneng Chen**[1,†]

[1]**JHU**  [2]**PKU**  [3]**Princeton**  [4]**MIT**  [5]**Harvard**
Project Page: *https://world-in-world.github.io/*

## ABSTRACT

Generative world models (WMs) can now simulate worlds with striking visual realism, which naturally raises the question of whether they can endow embodied agents with predictive perception for decision making. Progress on this question has been limited by fragmented evaluation: most existing benchmarks adopt open-loop protocols that emphasize *visual quality* in isolation, leaving the core issue of *embodied utility* unresolved, i.e., *do WMs actually help agents succeed at embodied tasks?* To address this gap, we introduce World-In-World, the first open platform that benchmarks WMs in a closed-loop world that mirrors real agent-environment interactions. World-In-World provides a unified online planning strategy and a standardized action API, enabling heterogeneous WMs for decision making. We curate four closed-loop environments that rigorously evaluate diverse WMs, prioritize task success as the primary metric, and move beyond the common focus on visual quality; we also present the first data scaling law for world models in embodied settings. Our study uncovers three surprises: (1) visual quality alone does not guarantee task success—controllability matters more; (2) scaling post-training with action-observation data is more effective than upgrading the pretrained video generators; and (3) allocating more inference-time compute allows WMs to substantially improve closed-loop performance. By centering evaluation on closed-loop outcomes, World-In-World establishes a new benchmark for the systematic assessment of WMs.

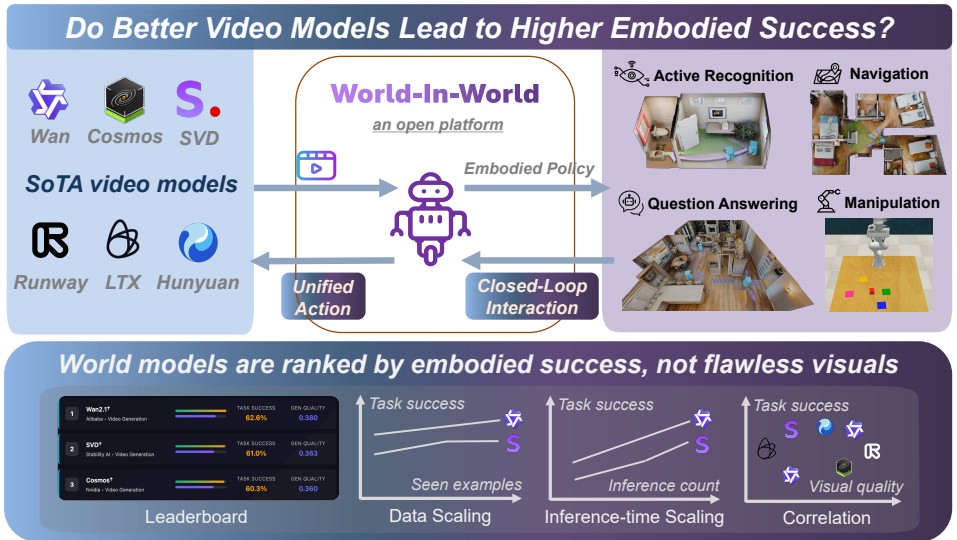

Figure 1: We introduce the first open benchmark to evaluate world models by closed-loop task success, analyze the link between task success and visual quality, and investigate scaling laws.

# 1 INTRODUCTION

Recent advances in visual generation have sparked interest in world generation, a field focused on the creation of diverse environments populated with varied scenes and entities, with applications in entertainment, gaming, simulation, and embodied AI. The rapid progress in video generation (Brooks et al., 2024; Yang et al., 2024b; Wan et al., 2025), 3D scene generation (Fridman et al., 2023; Chung et al., 2023; Yu et al., 2024; Koh et al., 2023; Ling et al., 2025), and 4D scene generation (Bahmani et al., 2024b; Xu et al., 2024; Bahmani et al., 2024a) has demonstrated high-quality individual scene generation, highlighting the potential of these models as world generation systems.

Building on these developments, recent world generation systems (Yang et al., 2023b; Parker-Holder & Fruchter, 2025; Li et al., 2025c; Ye et al., 2025; Lu et al., 2025; He et al., 2025c) show promise as world models for embodied agents. Given an agent's initial observation and a candidate action, such systems predict the resulting video, thereby estimating the future state of the environment. These action-conditioned simulators mirror human mental models by forecasting future states and can provide missing context under partial observability. As a result, they offer a pathway to improved decision-making for embodied tasks that rely on perception, planning, and control.

Despite this promise, the community lacks a unified benchmark that evaluates visual world models *through the lens of embodied interaction*. Existing suites emphasize video generation quality (e.g., VBench (Huang et al., 2024)) or visual plausibility (e.g., WorldModelBench (Li et al., 2025b)). The recent WorldScore (Duan et al., 2025) offers a unified assessment for models that take an image and a camera trajectory as input. However, *no current benchmark tests whether generated worlds actually enhance embodied reasoning and task performance*—for example, helping an agent perceive the environment, plan and execute actions, and replan based on new observations *within such a closed loop*. Establishing this evaluation framework is essential for tracking genuine progress across the rapidly expanding landscape of visual world models and embodied AI.

In this work, we address this gap by proposing World-In-World, which wraps generative World models In a closed-loop World interface to measure their practical utility for embodied agents. Specifically, we present a unified strategy for closed-loop online planning and a standardized action API to seamlessly integrate diverse world models into closed-loop tasks. The online planning strategy allows the agent to look ahead by anticipating environmental changes and task rewards before committing to an action. The standardized action API harmonizes input modalities expected by different world models, so that each model can be controlled consistently within the same evaluation protocol. In addition, we introduce a post-training protocol that fine-tunes pre-trained video generators using a modest amount of action–observation data drawn from the same action space as the downstream tasks, which allows us to examine their adaptation potential and to characterize a data scaling law.

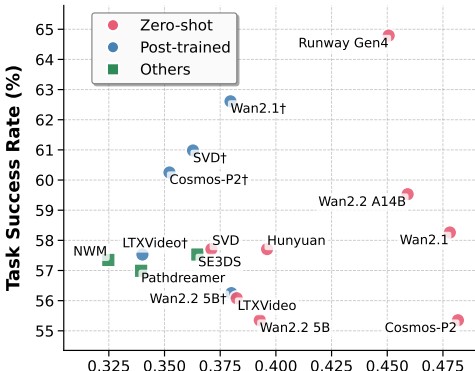

Figure 2: Task success rate vs. generation quality from VBench. †: post-trained with extra data. We defend that world models live and die by their closed-loop success, not flawless generated visuals.

World-In-World offers a fair, closed-loop world interface to evaluate diverse WMs. We benchmark leading video generators (Wan et al., 2025; HaCohen et al., 2024; Kong et al., 2024) alongside task-focused world models (Bar et al., 2025; Koh et al., 2023; 2021) in perception, navigation, and manipulation settings. Our findings reveal three consistent trends: (1) high visual quality does not necessarily translate into strong task success; (2) scaling post-training with action-observation data is more effective than upgrading the pretrained video generators; and (3) increasing inference-time compute via online planning substantially improves closed-loop performance. As shown in Figure 2, world models with strong visual scores do not necessarily bring high success rates, which underscores the need for closed-loop evaluation when judging WM practical value for embodied agents.

Our work makes three main contributions:

- We introduce World-In-World, the first comprehensive *closed-loop* benchmark that evaluates world models through the lens of embodied interaction, moving beyond the common focus on generation quality.

- We propose a *unified closed-loop planning* strategy with a *unified action API*, enabling diverse world models to be integrated and assessed within one framework across four embodied tasks.

- We discover that high visual quality does not necessarily guarantee task success, and demonstrate how the performance of pretrained video generators can be substantially improved through *training-time data scaling* and *inference-time scaling*.

## 2 WORLD-IN-WORLD: A CLOSED-LOOP INTERFACE FOR VISUAL WORLD MODELS

**Design overview**. Our goal is to establish a benchmark that evaluates world-generation methods by their utility for embodied agents. Unlike prior work focused on generative quality, we develop a predictive-control framework to test how well a world model supports online decision-making. The evaluation setting mirrors practical scenarios in embodied AI, emphasizing the interaction between prediction, control, and reward under closed-loop operation.

We detail the unified strategy for closed-loop online planning (Section 2.1) and the unified action API (Section 2.2), which together provide a common interface across tasks and models. We then describe our task selection and evaluation protocol (Section 2.3). Finally, we present a post-training recipe that adapts a pretrained video generator into a more effective embodied world model (Section 2.4).

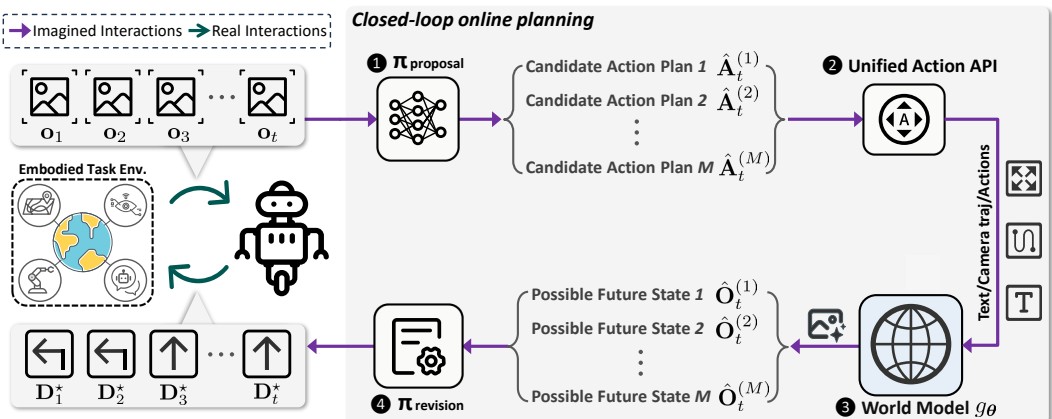

Figure 3: Closed-loop online planning in World-In-World: At time step $t$, the agent receives the world state, represented by observation $\mathbf{o}_t$, and invokes a proposal policy $\pi_{\text{proposal}}$ (❶) to produce a total of $M$ candidate action plans. The unified action API (❷) transforms each plan into the control inputs required by the world model. The world model (❸) then predicts the corresponding future states as observations $\hat{\mathbf{O}}_t$. The revision policy $\pi_{\text{revision}}$ (❹) evaluates all rollouts and commits to the best, yielding decision $\mathbf{D}_t^\star$. This decision is applied in the environment, closing the interaction loop.

### 2.1 UNIFIED STRATEGY FOR CLOSED-LOOP ONLINE PLANNING

In Figure 3, we present a unified closed-loop strategy that uses visual world models for decision-making. It cycles through *proposal*, *simulation*, and *revision*. In *proposal*, the agent generates candidate plans; in *simulation*, each plan is rolled out by the world model to predict counterfactual futures; in *revision*, the agent scores rollouts and refines its plan. Finally, the agent executes the top-scoring plan in the environment, coupling model-based planning with real execution.

Let $\mathbf{o}_t$ denote the agent's egocentric observation at time step $t$.[1] Define the agent's future potential action sequence of horizon $L$ starting at time step $t$ as $\hat{\mathbf{A}}_t = \left[\hat{a}_{t+1}, \hat{a}_{t+2}, \ldots, \hat{a}_{t+L}\right]$, where each

---

[1]The observation may be RGB, RGB-D, or another sensory modality. For clarity, we use $\mathbf{o}$ as the generic notation throughout.

elementary action $\hat{a}$ is specified in either a continuous action space or a discrete action space, i.e., $\hat{a} \in \mathcal{V}$, with $\mathcal{V}$ denoting the set of action primitives available to the agent.

Our unified strategy can be formalized as a policy-guided beam search. The beam width corresponds to the number of candidate plans $M$ drawn from the proposal policy $\pi_{\text{proposal}}$. At time step $t$, given the current observation $\mathbf{o}_t$ and the task goal g, the proposal policy $\pi_{\text{proposal}}$ samples $M$ candidate action sequences that serve as future candidate plans:

$$\hat{\mathbf{A}}_t^{(m)} \sim \pi_{\text{proposal}}\left(\mathbf{A} \mid \mathbf{o}_t, \text{g}\right), \qquad m = 1, \ldots, M. \tag{1}$$

Each candidate plan $\hat{\mathbf{A}}_t^{(m)}$ is subsequently transformed by the unified action API $C$ into the control inputs expected by the world model: $I_t^{(m)} = C\left(\hat{\mathbf{A}}_t^{(m)}\right)$, where $I_t^{(m)}$ may include textual prompts, camera trajectories, or low-level action sequences, depending on the required format of the chosen world model. The visual world model $g_{\boldsymbol{\theta}}$ then performs a counterfactual rollout based on these control inputs, predicting the future world states $\hat{\mathbf{O}}_t^{(m)}$ with horizon $L$:

$$\hat{\mathbf{O}}_t^{(m)} \sim g_{\boldsymbol{\theta}}\left(\mathbf{O} \mid \mathbf{o}_t, I_t^{(m)}\right), \qquad \hat{\mathbf{O}}_t^{(m)} = \left[\hat{\mathbf{o}}_{t+1}^{(m)}, \hat{\mathbf{o}}_{t+2}^{(m)}, \ldots, \hat{\mathbf{o}}_{t+L}^{(m)}\right]. \tag{2}$$

Then, the candidate plans and their simulated rollouts $\left(\hat{\mathbf{A}}_t^{(m)}, \hat{\mathbf{O}}_t^{(m)}\right)$ are evaluated and revised by the revision policy $\pi_{\text{revision}}$, which assigns a score to each trajectory and selects the decision that maximizes the expected reward. In the most general form, we write

$$\mathbf{D}_t^{\star} = \pi_{\text{revision}}\left(\left\{\left(\hat{\mathbf{A}}_t^{(m)}, \hat{\mathbf{O}}_t^{(m)}\right)\right\}_{m=1}^{M}, \mathbf{o}_t, \text{g}\right). \tag{3}$$

Here, $\mathbf{D}_t^{\star}$ denotes the best decision according to $\pi_{\text{revision}}$ at time step $t$. Depending on the task, $\mathbf{D}_t^{\star}$ may represent a high-level answer, a recognition result, or a refined sequence of low-level actions, which renders the framework more general than classical Model Predictive Control (MPC) (Morari & H. Lee, 1999), where optimization is typically restricted to sequences of actions.

A common instantiation implements $\pi_{\text{revision}}$ as a score-and-select operator $S$. When the decision is an action sequence, selection is performed over the $M$ candidate plans produced at time step $t$:

$$\mathbf{D}_t^{\star} = \hat{\mathbf{A}}_t^{(m^{\star})}, \quad \text{where} \quad m^{\star} = \underset{m \in \{1, \ldots, M\}}{\arg\max} \ S\left(\hat{\mathbf{A}}_t^{(m)}, \hat{\mathbf{O}}_t^{(m)} \mid \mathbf{o}_t, \text{g}\right). \tag{4}$$

Here, $S(\cdot)$ denotes a task-specific scoring function that estimates the expected reward or utility of a candidate plan based on its simulated outcomes. Alternatively, $\pi_{\text{revision}}$ may synthesize or update a new decision by aggregating information across the candidate set and their predicted consequences, rather than selecting one candidate verbatim.

Once the best decision $\mathbf{D}_t^{\star}$ is executed in the environment, the agent acquires a new observation at time step $t+1$. The unified strategy then re-enters the proposal-simulation-revision loop, using the newly observed state to initiate the next round of proposal, simulation, and revision. In our framework, both $\pi_{\text{proposal}}$ and $\pi_{\text{revision}}$ can be instantiated flexibly: they may be pretrained modules, such as large-scale vision-language models or diffusion policies, or simple rule-based heuristics. In our experiments, we explore multiple instantiations to systematically explore the flexibility and generality of our framework for different tasks.

## 2.2 UNIFIED ACTION API

In this section, we present a unified action API that transforms an action sequence $\mathbf{A}$ into control inputs $I$ that guide the world model, i.e., $I = C(\mathbf{A})$. The action API is designed to be flexible so that the same interface can serve a wide range of world models and tasks. It supports three principal types of control information: (1) text prompt, (2) camera trajectory/viewpoint, and (3) low-level actions, depending on the inputs expected by the chosen world model.

**Text prompt.** For image-and-text-to-video world models, the controller maps the intended action sequence into a descriptive text prompt. A predefined template converts each primitive action into a phrase, and concatenating these phrases yields the final prompt $I_{\text{text}}$.

**Camera trajectory / viewpoint.** For models that consume explicit viewpoints, the controller translates $\mathbf{A}$ into a camera trajectory, e.g., each translation action moves the camera by $0.2\,\text{m}$, and

each rotation action changes the azimuth by $22.5°$. The resulting trajectory is represented as a sequence $\left[(x_k, y_k, \phi_k)\right]_{k=1}^{K}$ with $(x_k, y_k) \in \mathbb{R}^2$ and azimuth $\phi_k \in \mathbb{R}$.

**Low-level actions.** For world models that take discrete or continuous low-level actions as input, the controller maps the action sequence $\mathbf{A}$ to the world model's action vocabulary, yielding $\mathbf{A}_{\text{world}}$. This mapping $\mathbf{A} \mapsto \mathbf{A}_{\text{world}}$ applies the necessary transformations to maintain a unique and consistent correspondence between the agent's actions and the inputs expected by the world model.

## 2.3 COMPREHENSIVE EMBODIED TASKS

To evaluate the practical utility of visual world models in embodied tasks, we select a diverse set of tasks that span multiple domains and stress distinct capabilities. We focus on four representative tasks: *Active Recognition* (AR), *Active Embodied Question Answering* (A-EQA), *Image-Goal Navigation* (ImageNav), and *Robotic Manipulation*, as illustrated in Figure 4. Taken together, these tasks emphasize complementary aspects of embodied intelligence, including perception, navigation, and object-level manipulation, and thus provide a comprehensive testbed for assessing how effectively a visual world model supports online planning and decision-making. Below, we describe the tasks included in our benchmark, and more detailed settings are provided in Appendix B.

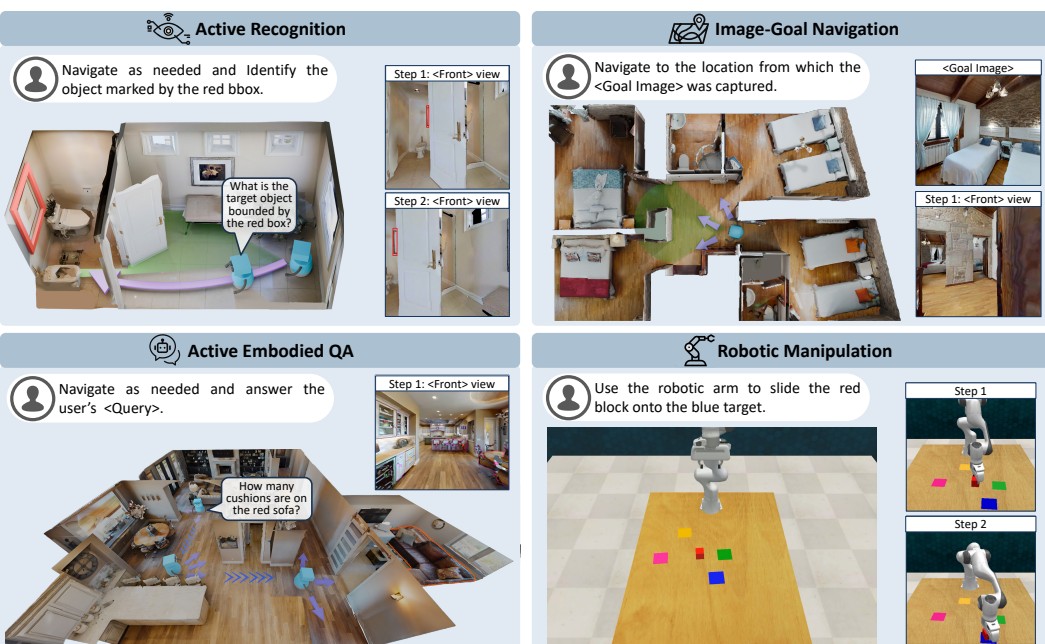

Figure 4: **Top-left**: Active Recognition (AR), the agent needs to identify a designated target under occlusions or extreme viewpoints while minimizing navigation cost. **Top-right**: Image-Goal Navigation (ImageNav), the agent reaches the viewpoint matching a goal image, emphasizing success rate and path efficiency. **Bottom-left**: Active Embodied Question Answering (A-EQA), the agent answers an open-ended question after active exploration. **Bottom-right**: Robotic Manipulation, the agent needs to control a robotic arm to complete tasks such as grasping and placement to specified targets.

**Active Recognition (AR)** is closely related to amodal recognition (Aydemir et al., 2013; Liu et al., 2018; Yang et al., 2019; Fan et al., 2024; Bhattacharjee et al., 2025), in which the agent must identify a designated target that may be observed from extreme viewpoints or be heavily occluded. In addition, AR allows the agent to acquire additional observations through active exploration. All AR experiments are conducted in the Habitat-Sim (Savva et al., 2019), encompassing 551 episodes across 29 scenes from the validation split of Matterport3D (Chang et al., 2017). Within AR, the visual world model assists two decision-making processes. For answering, synthetic views provide auxiliary evidence that helps the agent reason about occlusions and extreme viewpoints that impede recognition. For navigation, rollouts simulate the consequences of potential actions so that the agent can choose a path that is more likely to yield informative observations.

**Image-Goal Navigation (ImageNav)**, also referred to as goal-conditioned visual navigation, requires an embodied agent to reach a target position in a scene given a single reference image that specifies the goal viewpoint. We construct 144 ImageNav episodes from 87 validation scenes of HM3D (Ramakrishnan et al., 2021). In this task, the visual world model exclusively supports navigation decisions. The agent simulates the outcomes of candidate action plans, selects the best option, executes the first segment of that plan, and then replans with the newly observed state in a closed-loop manner.

**Active Embodied Question Answering (A-EQA)** requires an agent to answer open-ended natural-language questions after actively exploring a 3D environment. Our evaluation set includes 184 questions across 54 indoor scenes from the official OpenEQA split (Majumdar et al., 2024) and the HM3D validation set (Ramakrishnan et al., 2021). As in AR, the visual world model supports both question answering and navigation. For answering, synthetic views generated by the world model provide complementary perspectives that help resolve references to occluded or distant objects. For navigation, the agent simulates high-level action plans using the world model's predictions to choose exploration strategies likely to reveal question-relevant information.

**Robotic Manipulations** are fundamental capabilities for embodied agents that must operate in real-world interaction settings. We study how visual world models contribute to closed-loop manipulation planning, evaluating performance on four RLBench (James et al., 2020) tasks with 50 episodes per task. In our setting, the visual world model supports the agent in assessing candidate 7-DoF gripper actions by providing visual evidence about anticipated object motions and interactions, which enables a comparison of alternative plans before execution. The predicted outcomes then guide the selection of actions that are more likely to achieve the specified objective, thereby linking visual prediction accuracy to improvements in manipulation performance.

## 2.4 EXPLOITING WORLD MODELS VIA POST-TRAINING

To evaluate the feasibility of adapting pretrained video generators for embodied tasks, we introduce a post-training procedure that aligns a pretrained model with the domain distribution and action space of target environments. We perform fine-tuning separately on data from two simulators, Habitat-Sim and CoppeliaSim, to match the corresponding task domains. For Habitat-Sim tasks (AR, A-EQA, ImageNav), we post-train on a panoramic action-observation dataset collected from the HM3D (Ramakrishnan et al., 2021) training split. For CoppeliaSim tasks (Robotic Manipulation), we post-train on task demonstrations generated with RLBench (James et al., 2020). To assess generalization rather than memorization, all Habitat-Sim data used for post-training are sourced from scenes that are disjoint from our evaluation scenes, so the scenes in our evaluation tasks remain *unseen* by the world models after post-training. Additional details regarding the training objective, dataset construction, and training configuration are provided in Appendices C and D.

## 3 EVALUATION RESULTS AND ANALYSIS

In this section, we report quantitative results and key observations on the four embodied tasks in Section 3.1, followed by ablation studies in Section 3.2. We evaluate visual world models spanning image-based (PathDreamer (Koh et al., 2021), SE3DS (Koh et al., 2023)) and video-based (SVD (Blattmann et al., 2023a), LTX-Video (HaCohen et al., 2024), Hunyuan (Kong et al., 2024), Wan2.1 (Wan et al., 2025), Wan2.2 (Wan et al., 2025), Cosmos-Predict2 (Agarwal et al., 2025), NWM (Bar et al., 2025)) approaches, covering major control interfaces. For video-based models, we compare off-the-shelf versions with their post-trained variants, where the additional postfix "†" denotes a post-trained video generator.

## 3.1 BENCHMARK RESULTS

**World models can enhance the performance of the base proposal policy.** Across AR, A-EQA, ImageNav, and Manipulation, adding a visual world model consistently improves the performance of the base proposal policy (e.g., a VLM policy, a heuristic policy, or a 3D diffusion policy), as shown in Tables 1 to 3. For example, in AR, the best proprietary model (Runway Gen4) attains an accuracy of 64.79% while reducing the mean steps per episode to 4.06, compared to the VLM base policy with an accuracy of 50.27% and mean steps 6.24. Similarly, in ImageNav, the best open-source model Wan2.1† achieves a success rate of 45.14% with an average path length of 45.8, outperforming the VLM base policy at 35.42% SR and 47.5 average length. In A-EQA, the top post-trained model

Table 1: Active Recognition (AR) and Image-Goal Navigation (ImageNav) performance across various models and base policies. Higher success rate (**SR**%), success weighted by path length (**SPL**%), and lower mean trajectory length (**Mean Traj.**) are better. "†" denotes our post-trained video generators. "A14B" denotes a mixture-of-experts configuration of Wan2.2 with an effective model size of 14B during inference.

| Model Details | | | | | AR | | ImageNav | | |
|---|---|---|---|---|---|---|---|---|---|
| Model Type | Method | Control Type | Input Type | #Param. | SR↑ | Mean Traj.↓ | SR↑ | Mean Traj.↓ | SPL↑ |
| Base Policy | Heuristic (w/o WM) | – | RGB | – | 39.02 | 8.81 | 2.08 | 59.6 | 0.63 |
| + Video Gen. Post-Train | SVD† | Action | RGB; Pano | 1.5B | 60.62 | 5.17 | 20.83 | 58.5 | 11.86 |
| | WAN2.1† | Action | RGB; Pano | 14B | 62.98 | 4.71 | 22.92 | 58.7 | 11.63 |
| Base Policy | VLM (w/o WM) | – | RGB | 72B | 50.27 | 6.24 | 35.42 | 47.5 | 25.88 |
| + Image Gen. | PathDreamer | Viewpoint | RGB-D; Pano | 0.69B | 56.99 | 5.28 | 36.80 | 47.3 | 26.85 |
| + Image Gen. | SE3DS | Viewpoint | RGB-D; Pano | 1.1B | 57.53 | 5.29 | 36.11 | 47.0 | 26.91 |
| + Video Gen. | NWM | Trajectory | RGB | 1B | 57.35 | 5.68 | 40.28 | 47.1 | 27.83 |
| + Video Gen. Zero-Shot | SVD | Image | RGB | 1.5B | 57.71 | 5.29 | 40.28 | 46.4 | 28.59 |
| | LTX-Video | Text | RGB | 2B | 56.08 | 5.37 | 36.81 | 47.5 | 25.85 |
| | Hunyuan | Text | RGB | 13B | 57.71 | 5.21 | 36.11 | 46.8 | 26.89 |
| | Wan2.1 | Text | RGB | 14B | 58.26 | 5.24 | 38.19 | 48.2 | 25.92 |
| | Wan2.2 | Text | RGB | 5B | 55.35 | 5.73 | 38.88 | 46.5 | 28.87 |
| | Cosmos-P2 | Text | RGB | 2B | 55.35 | 5.71 | 36.81 | 47.6 | 25.89 |
| | Cosmos-P2.5 | Text | RGB | 2B | 58.26 | 5.12 | 36.81 | 47.7 | 26.57 |
| | Wan2.2 | Text | RGB | A14B | **59.53** | **4.91** | 43.05 | 45.8 | 31.46 |
| | Runway Gen4 (proprietary) | Text | RGB | – | 64.79 | 4.06 | - | - | - |
| + Video Gen. Post-Train | SVD† | Action | RGB; Pano | 1.5B | 60.98 | 5.02 | 43.05 | 46.0 | 30.96 |
| | LTX-Video† | Action | RGB; Pano | 2B | 57.53 | 5.49 | 38.89 | 47.4 | 27.47 |
| | WAN2.1† | Action | RGB; Pano | 14B | **62.61** | 4.73 | 45.14 | 45.8 | 32.10 |
| | Cosmos-P2† | Action | RGB; Pano | 2B | 60.25 | 5.08 | 41.67 | 45.5 | 30.29 |
| | Wan2.2† | Action | RGB; Pano | 5B | 56.26 | 5.15 | 38.89 | 46.7 | 28.24 |
| | Wan2.2† | Action | RGB; Pano | A14B | 62.43 | **4.67** | **46.53** | **44.6** | **34.61** |

Table 2: Active Embodied Question Answering (A-EQA) performance.

| Model Details | | A-EQA Performance | | |
|---|---|---|---|---|
| Model Type | Method | Ans. Score↑ | Mean Traj.↓ | SPL↑ |
| Base Policy | VLM (w/o WM) | 45.7 | 20.4 | 29.6 |
| + Image Gen. | PathDreamer | 46.0 | 20.4 | 29.3 |
| + Image Gen. | SE3DS | 45.8 | 20.3 | 29.4 |
| + Video Gen. | NWM | 47.1 | 20.5 | 30.1 |
| + Video Gen. | Wan2.1 | 45.7 | **20.1** | 28.8 |
| | Wan2.2 (5B) | 46.3 | 20.3 | 31.4 |
| | LTX-Video | 46.6 | 20.8 | 29.5 |
| | Cosmos-P2 | 46.6 | 21.0 | 31.3 |
| | Hunyuan | 46.8 | 20.4 | 29.9 |
| | SVD | 46.9 | 20.4 | 29.7 |
| | Wan2.2 (A14B) | **47.2** | 20.7 | **31.9** |
| + Video Gen. Post-Train | SVD† | 46.4 | 21.1 | 30.1 |
| | Cosmos-P2† | 46.5 | 20.6 | 30.1 |
| | Wan2.2† (5B) | 47.5 | 20.8 | 30.7 |
| | Wan2.1† | 48.2 | 20.7 | 31.6 |
| | LTX-Video† | **48.6** | 20.7 | 31.8 |
| | Wan2.2† (A14B) | 48.4 | **20.2** | **31.9** |

Table 3: Robotic manipulation performance across various models and base policies.

| Model Details | | Manipulation Performance | |
|---|---|---|---|
| Model Type | Method | SR↑ | Mean Traj.↓ |
| Base Policy | VLM (w/o WM) | 44.5 | 2.52 |
| + Video Gen. | SVD | 44.0 | 2.47 |
| | LTX-Video | 44.5 | 2.46 |
| | Hunyuan | 44.5 | 2.44 |
| | Wan2.1 | 44.0 | 2.51 |
| | Cosmos-P2 | 44.0 | 2.50 |
| + Video Gen. Post-Train | SVD† | 46.5 | 2.38 |
| | Cosmos-P2† | 45.0 | 2.40 |
| Base Policy | 3D-DP (w/o WM) | 24.0 | 5.21 |
| + Video Gen. Post-Train | SVD† | 44.7 | 4.41 |
| | Cosmos-P2† | 38.0 | 4.79 |

Wan2.2† A14B reaches an answer score of 48.4 and SPL of 31.9, surpassing the VLM base policy at 45.7 answer score and 29.6 SPL. These results support the effectiveness of our World-In-World online planning framework with world models, in which the world model provides simulated future states that inform better decisions.

**World models struggle to simulate precise motion and dynamics in manipulation.** The gains are less pronounced for Robotic Manipulations (Table 3), likely because accurately modeling contact-rich interactions and robot kinematics is significantly more challenging than predicting purely view changes. For instance, the best post-trained model on manipulation (SVD†) reaches an SR of 46.5% with a mean trajectory length of 2.38, only modestly above the VLM baseline at 44.5% SR and 2.52 mean length. This gap suggests that while current visual world models can effectively guide perception and navigation, capturing fine-grained physical dynamics and action-conditioned object motion remains an open challenge.

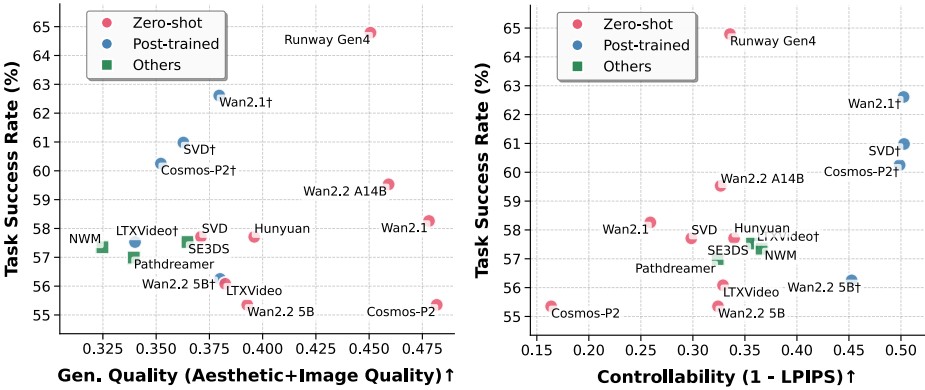

Figure 5: **(a)** SR vs. generation quality in AR; generation quality is scored as the average of an aesthetic predictor (Akio Kodaira, 2024) and an image-quality predictor (Ke et al., 2021), both trained to match human preferences. **(b)** SR vs. controllability in AR; controllability is quantified as $1 - $ LPIPS between ground-truth and predicted observations.

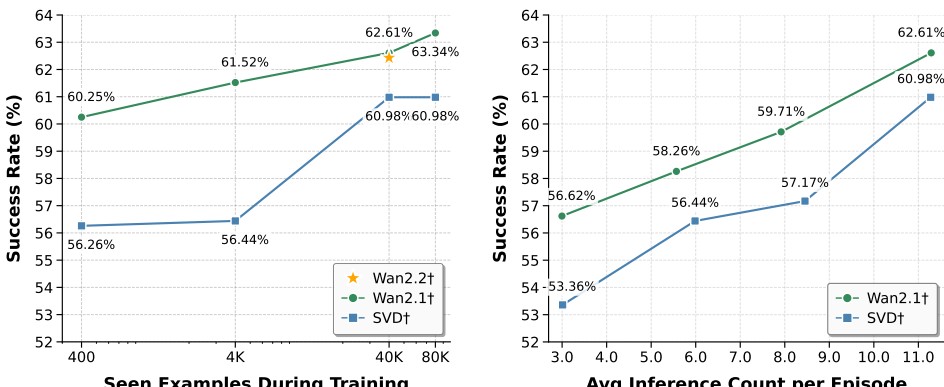

Figure 6: SR vs. seen examples during post-training. SR increases consistently with more downstream data, revealing a clear data-scaling trend for adaptation.

Figure 7: SR vs. average number of world-model inferences per episode. Increasing the inference-time computation allocated to each decision step leads to higher SR.

**Post-training substantially boosts world-model utility.** Our post-training adaptation yields consistent improvements. Relative to off-the-shelf Wan2.1, Wan2.1† raises AR accuracy from $58.26\%$ to $62.61\%$ and ImageNav SR from $38.19\%$ to $45.14\%$ (Table 1). Likewise, SVD† improves AR accuracy from $57.71\%$ to $60.98\%$ and ImageNav SR from $40.28\%$ to $43.05\%$. In A-EQA, LTX-Video† increases the answer score from $46.6$ to $48.6$, and Wan2.1† from $45.7$ to $48.2$. These gains show that aligning the generative model to the target domain and action space of the specific embodied tasks improves downstream decision-making.

## 3.2 ABLATION AND FINDINGS

**Fine-grained controllability matters more than visuals for task success.** Although recent off-the-shelf video generators like Wan2.1 produce visually appealing clips, they are driven by text prompts with limited fine-grained low-level controls. Without adaptation, these models yield only small gains on downstream embodied tasks. We further study the relation between controllability and the success rate on AR. Here, controllability is defined as alignment between intended actions and the motions in the model's predictions. After action-conditioned post-training, alignment improves substantially and SR rises accordingly. Figure 5(b) shows a clearer positive correlation than Figure 5(a), which depicts SR versus generation quality (aesthetic and image-quality scores), and suggests that models that respond reliably to low-level controls achieve higher SR. These results indicate that precise control, not just visual quality, is critical for embodied world models to support effective decision-making.

**Data-size scaling for post-trained models.** We study how post-training data size affects WM performance (Wan2.2†, Wan2.1†, SVD†). Each WM is post-trained for one epoch on datasets from 400 to 80K instances. As shown in Figure 6, more post-training data consistently improves AR performance: Wan2.1† rises from 60.25% to 63.34%, and SVD† from 56.80% to 60.98%. Wan2.2† (A14B), despite substantially larger web-video pretraining, reaches nearly the same performance as Wan2.1† after 40K post-training instances, suggesting that scaling action-conditioned post-training is more effective for embodied utility than upgrading the pretrained generator. Moreover, larger models (Wan2.1†, 14B) benefit more and saturate less than smaller ones (SVD†, 1.5B), indicating greater capacity to absorb action-conditioned supervision.

**Inference-time scaling for online planning with world models.** Within our online planning framework, the number of world-model inferences (simulated potential futures per episode) directly affects task performance. As shown in Figure 7, increasing the average inferences per episode for AR yields a clear positive correlation with SR. For example, increasing the average inference count from 3 to 11 improves SR from 53.36% to 60.98% for SVD†. This suggests that allocating more inference-time computation to simulate potential futures lets the planner make more informed decisions, thereby improving overall performance.

**Global vs. local context for generation.** We study the effect of input context format. Specifically, we compare post-trained models conditioned on panoramic versus front-view input (Table 4). Panoramic input provides a 360° field of view, whereas front view offers a focused but limited perspective. For fairness, generated panoramas are converted to perspective views with the same horizontal field of view during evaluation. Although panoramic input offers richer global context, it does not consistently yield large gains across all settings. Likely, panorama-to-perspective conversion introduces resolution loss, degrading downstream perception and planning.

Table 4: Post-training with different input contexts: front view vs. panorama.

| Task | Model | Front View | | Panorama | |
|------|-------|---|---|---|---|
| | | SR ↑ | Mean Traj. ↓ | SR ↑ | Mean Traj. ↓ |
| AR | SVD† | 57.89 | 5.04 | 60.98 | 5.02 |
| | Wan2.1† | 62.25 | 4.82 | 62.61 | 4.73 |
| | Wan2.2† (5B) | 57.16 | 5.08 | 56.26 | 5.15 |
| | Cosmos-P2† | 58.98 | 4.94 | 60.25 | 5.08 |
| ImageNav | SVD† | 38.19 | 47.0 | 43.05 | 46.0 |
| | Wan2.1† | 48.61 | 43.8 | 45.14 | 45.8 |
| | Wan2.2† (5B) | 40.97 | 45.8 | 38.89 | 46.7 |
| | Cosmos-P2† | 40.97 | 47.0 | 41.67 | 45.5 |

**Effect of different revision policies.** We study how the revision policy affects task performance by comparing a VLM-based revision policy with a simple LPIPS-based policy that selects the candidate whose predicted observation is closest to the goal image in perceptual feature space. From Table 5, we see that even a simple LPIPS-based revision policy could improve the performance significantly: SVD† obtains 47.92% SR and 39.82 SPL compared with 43.05% SR and 30.96 SPL using a VLM-based revision policy and 35.42% SR and 25.88 SPL without any WM augmentation. Augmenting the planner with action-conditioned WMs and applying a simple LPIPS-based revision can yield a higher SR and more efficient navigation.

Table 5: Effect of world-model augmentation and revision policy on ImageNav. SR and SPL are higher-is-better; mean trajectory length is lower-is-better.

| $\pi_{proposal}$ | WM Type | $\pi_{revision}$ | SR ↑ | Mean Traj. ↓ | SPL ↑ |
|------|---------|----------|------|-----------|-------|
| VLM | None | None | 35.42 | 47.5 | 25.88 |
| VLM | SVD† | VLM | 43.05 | 46.0 | 30.96 |
| VLM | Wan2.1† | VLM | 45.14 | 45.8 | 32.10 |
| VLM | SVD† | LPIPS | 47.92 | 41.3 | 39.82 |
| VLM | Wan2.1† | LPIPS | 48.61 | 39.8 | 42.48 |

**Domain transfer across scene distributions.** We evaluate cross-domain generalization by post-training WMs on the synthetic Habitat Synthetic Scenes Dataset (HSSD) and testing them on our AR and ImageNav suites built on the real-world scenes in HM3D/MP3D (Table 6). Despite the synthetic-to-real gap, HSSD-trained WMs still yield clear gains over the VLM-only baseline (e.g., SVD† improves AR SR from 50.27% to 58.98% and ImageNav SR from 35.42% to 38.89%). Performance remains below in-domain post-training on HM3D (SVD†: 60.98% AR SR, 43.05% ImageNav SR), as expected under a stronger distribution shift. These results indicate that post-training learns action-conditioned visual representations that transfer across scene distributions, consistent with prior work on adaptable world models (Gao et al., 2025).

Table 6: Cross-domain post-training: WMs post-trained on HSSD or HM3D and evaluated on HM3D/MP3D (val) for AR and ImageNav.

| WM Aug. | Post-Train Env. | AR | | ImageNav | |
|---------|-----------------|---|---|---|---|
| | | SR ↑ | Mean Traj. ↓ | SR ↑ | SPL ↑ |
| w/o WM | None | 50.27 | 6.24 | 35.42 | 25.88 |
| +SVD† | HSSD | 58.98 | 5.24 | 38.89 | 27.60 |
| +Wan2.1† | HSSD | 62.98 | 4.78 | 42.36 | 31.18 |
| +SVD† | HM3D (train) | 60.98 | 5.02 | 43.05 | 30.96 |
| +Wan2.1† | HM3D (train) | 62.61 | 4.73 | 45.14 | 32.10 |

## 4 DISCUSSION AND FUTURE DIRECTIONS

**Generalization capacity of world models is critical for practical use.** Most video generators are pretrained on web videos. In unseen embodied environments, they may revert to training priors or ignore action controls, yielding plausible but physically or semantically inconsistent rollouts (see Figures 13 and 14). These deviations mislead planning and reduce success. Larger models or more pretraining data can partly help, but robust generalization remains central. Future work should prioritize strategies and action representations to improve transfer to novel environments, such as unified action representations (Gao et al., 2025; Wang et al., 2025f; Zhi et al., 2025; Wang et al., 2025e) and curriculum or domain-specific data collection (Zhao et al., 2025).

**Long-horizon planning with world models remains challenging.** In our experiments, visual world models simulate short-term changes but struggle on long horizons due to limited mechanisms for accumulating spatiotemporal history. We attempted to alleviate this issue by replacing front-view inputs with panoramas to provide global context, but gains were inconsistent across models and tasks. Future work should better encode and retrieve long-term dependencies, e.g., spatial memory (Zhou et al., 2025b; Xiao et al., 2025; Li et al., 2025d; Yu et al., 2025a; Ren et al., 2025; Wang et al., 2025c) and episode-level memory (Cai et al., 2025; Guo et al., 2025), to maintain scene-level context and enable coherent planning over extended horizons.

**Precise modeling of interactions and dynamics remains difficult.** For manipulation, capturing contact-rich interactions, compliance, friction, and state changes of articulated or deformable objects is essential. Current visual world models often miss these details, producing rollouts that violate physics and degrade planning and control—consistent with our observations and prior analyses (Kang et al., 2024; Li et al., 2025a). Promising directions include physics-guided motion generation (Wang et al., 2025a; Zhang et al., 2025b; Akkerman et al., 2025), inferring or generating physical properties to inform action-conditioned predictions (Cao et al., 2025; Gillman et al., 2025; Zhang et al., 2024b), and physics-aware reinforcement post-training (Wu et al., 2025; Liu et al., 2025). Integrating such signals into conditioning pathways may improve fidelity when precise dynamics are required.

**Stronger proposal and revision policies set the performance floor.** The agent's overall performance depends on both world-model fidelity and the strength of the proposal and revision policies that select and refine decisions. While simulated rollouts improve decision-making, base policies must be effective to provide a reliable starting point, and strengthening them raises the ceiling. Future work could explore stronger policies (Geng et al., 2025; Kim et al., 2025), and integration strategies that deepen synergy between world models and decision-making (Neary et al., 2025), such as more human-aligned reward models (Wang et al., 2024; Seneviratne et al., 2025; Rocamonde et al., 2023; Zhang et al., 2024a; Wang et al., 2025d; Wu et al., 2025).

**Computational cost and efficiency remain practical concerns.** Incorporating world models into model-based planning introduces additional computational overhead because multiple future rollouts must be simulated at each decision step. Although our experiments show that allocating more inference-time computation to the world model improves task performance, this extra cost may be impractical in settings with strict real-time constraints or limited hardware resources. Future work should therefore investigate more efficient world-model architectures (Yang et al., 2025b; Kodaira et al., 2025), training and inference strategies that enable near real-time rollouts (Huang et al., 2025; Cui et al., 2025), and distillation techniques (Wang et al., 2025b; Agarwal et al., 2025) that reduce computational demands while preserving the predictive fidelity of world models.

## 5 CONCLUSION

We introduce World-In-World, a closed-loop world interface and benchmark that evaluates generative world models via embodied interaction rather than isolated visual metrics. By unifying heterogeneous controls, our action API enables any world model to serve as perception and planning utilities for an embodied agent. Coupled with a unified closed-loop planning strategy that proposes, simulates, and revises action plans, the benchmark measures agent performance on four demanding tasks. Our experiments reveal large gaps between visual metrics and task success, underscoring the need for closed-loop evaluation, and show that pretrained video generators improve with post-training data scaling and inference-time scaling. We expect World-In-World to guide world models toward not only striking visual realism but also reliable perception and planning in embodied scenarios.

**Acknowledgment** VP, RC, and YW were supported by the DARPA TIAMAT program.

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

# World-In-World: World Models in a Closed-Loop World

## Appendix

### CONTENTS

## A   RELATED WORK

**Visual generation.**   Recent advances in diffusion models (Sohl-Dickstein et al., 2015; Ho et al., 2020; Rombach et al., 2022; Brooks et al., 2024) have significantly improved the quality of image generation (Rombach et al., 2022; Zhang et al., 2023) and video generation (Blattmann et al., 2023b;a; Voleti et al., 2024; Xie et al., 2024), enabling temporally coherent and visually rich content synthesis from text prompts or a single image.  Image generators (Koh et al., 2021; 2023; Yu et al., 2023; Sargent et al., 2024; Seo et al., 2024) allow us to synthesize novel views with conditions on targeted viewpoints. Text-to-video generators such as Sora (Brooks et al., 2024) can generate minutes-long videos from text. Extensions incorporating camera trajectories as conditioning signals (Yin et al., 2023; Bar et al., 2025; He et al., 2025a;b; Zhou et al., 2025a; Bahmani et al., 2024a) push video generation toward dynamic scenes.  However, the absence of a unified conditioning framework hinders integration into downstream applications (*e.g.*, embodied decision making) and prevents fair cross-method comparisons. Moreover, these generative methods remain passive: generated worlds are treated as static backdrops and evaluated in an open-loop fashion using visual quality score (Huang et al., 2024) or controllability score (Duan et al., 2025).  In contrast, our work assesses not only generation quality but also closed-loop task success within a physical simulation.

**World models.**   Video-based generative models used as world models have shown effectiveness across a range of domains, including games (Alonso et al., 2024; Yu et al., 2025b; Li et al., 2025c; Ye et al., 2025; He et al., 2025c), manipulation (Du et al., 2023; Ko et al., 2023; Du et al., 2024; Yang et al., 2024a; Zhen et al., 2025), autonomous driving (Gao et al., 2024; Hu et al., 2023), and navigation (Bar et al., 2025; Wang et al., 2023; Koh et al., 2021), with extensions to broader embodied tasks (Lu et al., 2025; Zhang et al., 2025a; Long et al., 2025; Yang et al., 2025c). However, current evaluation frameworks for these world models are often limited to visual metrics (Duan et al., 2025; Li et al., 2025b) or to a single embodied task in a narrow domain (Bar et al., 2025; Zhen et al., 2025). The VP2 benchmark (Tian et al., 2023) moves toward a control-centric evaluation by measuring the utility of video prediction models in model-based planning. However, its simple setting, including limited task diversity and the use of earlier video prediction architectures, limits its relevance to modern video generative models and more complex embodied scenarios.  In contrast, our work provides a broader evaluation across four closed-loop embodied tasks, systematically benchmarking the practical utility of diverse world models that are pretrained on large-scale Internet video datasets.

## B   EMBODIED TASK DETAILS

This section details the setups for the four embodied tasks evaluated in World-In-World: Active Recognition (AR) in Appendix B.1, Image-Goal Navigation (ImageNav) in Appendix B.2, Active Embodied Question Answering (A-EQA) in Appendix B.3, and Robotic Manipulation in Appendix B.4. We also describe the policies used across these tasks in Appendix B.5 and summarize the world model details in Appendix B.6.

## B.1 ACTIVE RECOGNITION (AR)

All AR experiments are performed in Habitat-Sim using scenes from the validation split of Matterport3D (Chang et al., 2017). We focus on 29 scenes and curate a subset of 551 challenging episodes adapted from the dataset released by prior work (Fan et al., 2024). Each episode is manually inspected to ensure that it presents either an extreme viewpoint or a heavily occluded target object. These conditions force the agent to actively explore the environment and to rely on its world model for informed decision-making.

**Task setup.** In the AR setting, the agent is allowed at most $K = 10$ decision steps. At each step $t$, the agent receives an RGB observation $\mathbf{o}_t$ that includes a panoramic view and a front view with a horizontal field of view of $90°$. The agent's output at each step consists of answers to two multiple-choice queries: (i) which object category $\hat{y}_t$ matches the target. (ii) which navigation primitive $a_t \in \mathcal{V}$ to execute next. For each query, the VLM selects the token with the highest likelihood, and the associated probability is interpreted as the model's confidence. After choosing $a_t$, the agent executes the action, acquires the next observation, and proceeds to step $t+1$. The episode terminates when either the step budget $K$ is reached or the confidence of the predicted category $\hat{y}_t$ exceeds $95\%$.

**Integrating a world model.** Within the AR pipeline, the world model supports decision-making in two complementary ways that mirror the two queries above. For query (i), the model generates synthetic future views that act as auxiliary evidence in addition to the real observation $\mathbf{o}_t$. These additional cues help the agent reason about occlusions, extreme viewpoints, and other distribution shifts that hinder recognition, as illustrated in Figure 8. For query (ii), agent will first generate $M$ candidate action sequences $\{\mathbf{A}_t^m\}_{m=1}^M$, each of length $L$. Given each candidate plan and its corresponding predicted observations, the agent estimates the value of alternative low-level control sequences before committing to an action in the real environment. Unlike a baseline policy that greedily chooses $a_{t+1}$ from $\mathbf{o}_t$ alone, the agent equipped with a world model compares simulated outcomes for all candidates and executes the sequence that is expected to yield the most informative next view. When a world model is used, the planner proposes $M = 2$ candidate action sequences per step, each with horizon $L = 4$.

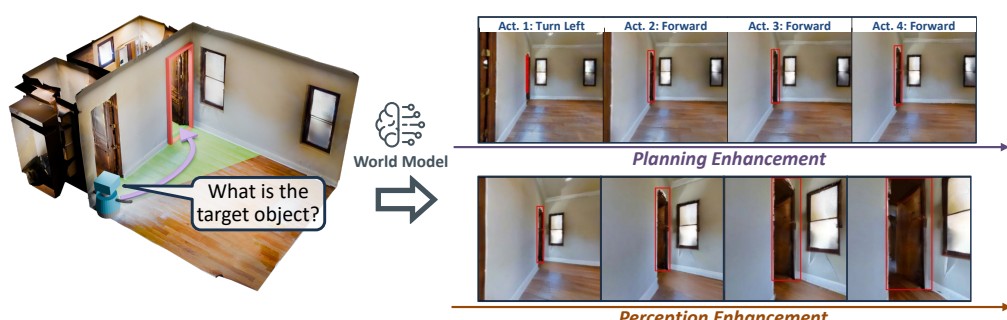

Figure 8: In AR, the world model supports both queries (perception and planning). In this example, the agent must identify a wooden door that is initially visible only from an extreme viewpoint. For each candidate action sequence, the world model predicts future observations; these forecasts augment the agent's perception and inform the choice of the next action.

**Bounding box annotation.** The target object is marked by a red bounding box overlaid on the image. For the current real observation $\mathbf{o}_t$, the box is obtained from Habitat ground-truth annotations. For the predicted frames $\{\hat{\mathbf{o}}_i\}_{i=t+1}^{t+L}$ produced by the world model, we apply SAM2 (Ravi et al., 2024) to segment the target, seeding the segmenter with the ground-truth box from the current real observation $\mathbf{o}_t$ to maintain correspondence across time.

**Metrics.** AR performance is reported using two metrics: (1) *Success Rate (SR)*, defined as the fraction of episodes in which the final predicted label $\hat{y}$ matches the ground-truth label $y$; and (2) *Mean Trajectory Length*, defined as the average number of executed actions before the agent either issues its final prediction or exhausts the step budget $K$.

## B.2 IMAGE-GOAL NAVIGATION (IMAGENAV)

Image-Goal Navigation (ImageNav), also known as goal-conditioned visual navigation, requires an embodied agent to reach the target location depicted by a single reference image of the goal. The environment is unknown, so the navigation policy must determine how to explore in order to locate the goal efficiently. To examine how world models can assist, we create 144 ImageNav episodes taken from 87 validation scenes of HM3D (Ramakrishnan et al., 2021).

**Task setup.** Each episode permits at most $K = 20$ decision steps. As in the AR setting, at step $t$ the agent receives an RGB observation $\mathbf{o}_t$ comprising a panoramic view and a front view with a horizontal field of view of $90°$. The agent then proposes a sequence of low-level navigation primitives $\mathbf{A}_t = [a_{t+1}, a_{t+2}, \ldots, a_{t+L}]$ with a maximum horizon of $L = 5$. The first $L - 2$ primitives from the selected plan are executed in the real environment, after which the agent replans based on the newly acquired observation. An episode is successful if, within the budget of $K$ steps, the agent's position enters a sphere of radius $R_g = 0.5$, m centered at the location specified by the goal image $\mathbf{g}$.

**Integrating a world model.** In ImageNav, the agent answers only the navigation query of which action sequence to execute next; therefore, the world model is used exclusively for *planning enhancement*. The agent first enumerates several candidate action sequences. For each candidate, the world model predicts the future observations that would follow if the sequence were executed from the current state. The agent then scores each sequence by assessing how informative its predictions are for locating the goal, and selects the sequence with the highest expected utility. When a world model is used, the planner proposes $M = 3$ candidate action sequences at each decision step, with horizon $L = 5$. The first $L - 2$ actions from the chosen sequence are carried out before the next cycle begins.

**Metrics.** We report three standard metrics for ImageNav: (1) *Success Rate (SR)*, the fraction of episodes in which the agent reaches the goal within the decision budget; (2) *Mean Trajectory Length*, the average number of executed actions across all episodes; and (3) *Success weighted by Path Length (SPL)*, which accounts for both success and path efficiency. Formally, for a set of $N$ episodes,

$$\text{SPL} = \frac{1}{N} \sum_{i=1}^{N} S_i \frac{L_i^*}{\max(L_i, L_i^*)} \times 100\%,$$

where $S_i \in \{0, 1\}$ indicates whether episode $i$ is successful, $L_i^*$ is the shortest path length from the start position to the goal for episode $i$, and $L_i$ is the actual path length executed by the agent in that episode.

## B.3 ACTIVE EMBODIED QUESTION ANSWERING (A-EQA)

Active Embodied Question Answering (A-EQA) tasks an embodied agent with answering open-ended, natural-language questions after actively exploring an environment. The questions span six broad categories that are common in embodied QA: recognizing objects, recognizing object attributes, recognizing object states, localizing objects, performing spatial reasoning, and performing functional reasoning. Our evaluation set contains 184 questions distributed across 54 indoor scenes drawn from the official OpenEQA split (Majumdar et al., 2024) and the validation set of HM3D (Ramakrishnan et al., 2021).

**Task setup.** In A-EQA, there is no predefined navigation goal, so the agent must design its own exploration strategy to gather sufficient visual evidence for answering the question. At every decision step $t$, the agent receives a panoramic RGB observation that we decompose into four perspective views, each with a horizontal field of view of $105°$ (see Figure 10). The exploration budget is limited to 250 low-level actions; a single decision step can comprise multiple low-level actions, depending on the high-level intent. An episode terminates when the budget is exhausted or when the agent outputs a final answer $\hat{y}$.

For A-EQA, we implement a two-level policy that separates deliberation and control. The high-level planner periodically issues one of two types of commands: (i) a textual instruction (for example, "move to the hallway visible in the front view"), or (ii) the index of a landmark object detected in the current panorama. Once a high-level command is produced, execution is delegated to the low-level controller. If the command specifies a landmark, the controller uses depth data together with a custom pathfinder to plan and follow a route to that landmark. If the command is a textual

instruction, the controller generates a sequence of low-level actions to carry out the instruction. This planner-controller loop continues until either the 250 atomic actions are consumed or the high-level planner decides to emit the final answer $\hat{y}$.

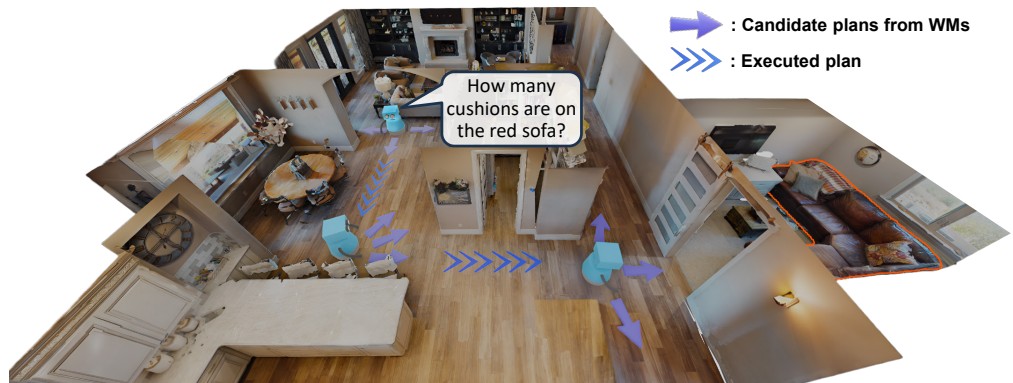

Figure 9: Overview of our embodied closed-loop evaluation for A-EQA. For each question, the high-level planner proposes multiple candidate action plans and queries the world model to generate the corresponding future observations. The agent then evaluates each plan together with its predicted observations and selects the plan that maximizes the expected reward before executing it in the environment.

**Integrating a world model.** In A-EQA, the world model is primarily used to strengthen the high-level planner. At each high-level decision point, the planner samples $M$ candidate action plans and queries the world model to produce the corresponding predicted observations, as illustrated in Figure 9. The agent then evaluates each plan-observation pair $(\hat{\mathbf{A}}_t^{(m)}, \hat{\mathbf{O}}_t^{(m)})$ and chooses the plan that maximizes the estimated reward under the current question context. This differs from the AR setting, where perception and planning are evaluated through two separate queries. In A-EQA, the high-level planner must both design a long-horizon exploration sequence *and* decide when to stop exploring to output a final answer $\hat{y}$. Consequently, the world model supports a single unified query: the predicted observations simultaneously refine the agent's understanding of the scene and provide forecasts for scoring alternative exploration plans. When a world model is enabled, the planner proposes $M = 3$ candidate sequences per step, each with horizon $L = 14$. Unlike AR or ImageNav, only the terminal predicted observation at step $L$ is returned to the high-level planner for scoring, rather than the full rollout over all $L$ steps.

**Landmark detection and labeling.** Landmark objects are detected by first running YOLO-World to obtain bounding boxes and then applying SAM2 to derive instance masks (Ravi et al., 2024; Cheng et al., 2024). This detection pipeline follows the Set-of-Marks (SoM) strategy (Yang et al., 2023a) shown in Figure 10 and provides a discrete set of navigable targets for high-level planning.

**Metrics.** A-EQA performance is evaluated with three metrics. (1) *Answering Score*: a large language model (e.g., GPT-4o) compares the agent's final answer $\hat{y}$ to the ground-truth answer $y$ and assigns a raw score in $[1, 5]$, where 5 indicates a perfect match. We average the raw score across episodes and then linearly map it to $[0, 100]$. (2) *Mean Trajectory Length*. This is the average travel distance the agent covers before either producing its final answer or exhausting the step budget $K$, lower is better. (3) *Success weighted by Path Length (SPL)*: this metric rewards both answer quality and navigation efficiency. For episodes in which the agent fails to return an answer, we fall back to its

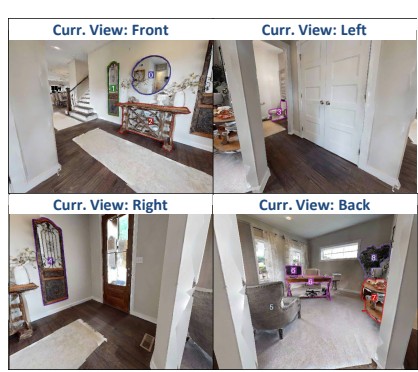

Figure 10: Illustration of the *Set-of-Marks* (SoM) representation that encodes candidate navigable directions. The high-level planner chooses among these discrete landmarks when constructing candidate action plans.

blind LLM variant and set the SPL contribution to zero. Formally,

$$\text{SPL}_{\text{A-EQA}} = \frac{1}{N} \sum_{i=1}^{N} \left( \frac{\sigma_i - 1}{4} \right) \frac{L_i^*}{\max(L_i, L_i^*)} \times 100\%,$$

where $N$ is the number of evaluation episodes, $\sigma_i \in [1, 5]$ denotes the raw Answering Score for episode $i$, $L_i^*$ denotes the shortest-path length from the start to a viewpoint that affords a correct answer, and $L_i$ denotes the actual path length executed by the agent in episode $i$. A higher value indicates both more accurate answering and more efficient exploration.

### B.4 ROBOTIC MANIPULATION

We study whether world models can improve low-level manipulation, which is a core capability for embodied agents. Our evaluation covers four robotic manipulation tasks in RLBench (James et al., 2020): Push Buttons, Slide Block to Color Target, Insert onto Square Peg, and Stack Cups. RLBench is a widely used benchmark for robot learning. Each episode provides a natural-language instruction that specifies the task objective, and the agent must control a 7-DoF robotic arm to satisfy that objective. We prepare a total of 200 evaluation episodes, with 50 episodes for each task.

**Task setup.** At each decision step $t$, the agent receives an observation $\mathbf{o}_t$ and proposes an action sequence $\mathbf{A}_t = \left[ \mathbf{a}_{t+1}, \mathbf{a}_{t+2}, \ldots, \mathbf{a}_{t+L} \right]$, where each low-level action is parameterized as $\mathbf{a}_t = [x, y, z, \text{roll}, \text{pitch}, \text{yaw}, \text{gripper}]$. We consider two base policy settings with different horizons: $L = 5$ for a VLM base policy that emits discrete actions, and $L = 50$ for a 3D diffusion base policy that emits continuous actions. An episode is counted as a success if the specified goal $\mathbf{g}$ is achieved within the step budget $K$.

When a VLM is the base policy, directly producing precise low-level controls is challenging for current VLMs. Following (Yang et al., 2025a), we therefore introduce two enhancements. First, we discretize the action space by dividing the position components $(x, y, z)$ into 100 bins and the orientation components (roll, pitch, yaw) into 120 bins. Second, we augment the observations with object index markers and provide precise object poses for indexed objects so that the VLM can directly access spatial information during planning (shown in Figure 11). Under this configuration, the manipulation policy is allowed at most $K = 15$ low-level action steps per episode. In contrast, when using a 3D diffusion policy (Ke et al., 2024) as the base policy, the controller naturally generates continuous low-level actions, so we do not apply the discretization or the additional indexing enhancements. In this configuration, the manipulation policy is permitted at most $K = 8$ macro decision steps per episode.

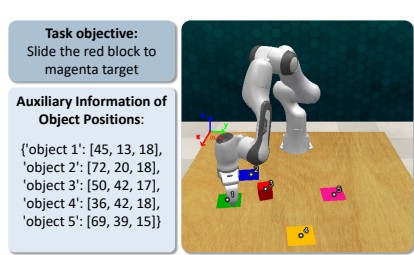

Figure 11: Illustration of the auxiliary information provided to the VLM policy. The objects are marked with indices, and their positions are given to the VLM to facilitate decision-making.

**Integrating a world model.** As in ImageNav, we use the world model exclusively for *planning enhancement*. The agent executes a propose, simulate, and revise loop so that it can reason about the consequences of alternative plans before applying any action in the real environment. At each decision step, the planner proposes $M = 5$ candidate action sequences. When the length of a candidate sequence is shorter than the world model's required action-conditioning length, the unified action API linearly interpolates the sequence to the required length. Conversely, when the candidate sequence is longer than required, the unified action API uniformly samples actions along the sequence to match the world model's input length. The planner then evaluates the simulated outcomes and selects the sequence with the highest expected reward, and the loop repeats with updated observations.

**Metrics.** We report two standard metrics for manipulation tasks: (1) *Success Rate (SR)*, the fraction of episodes in which the agent reaches the goal within the decision budget; and (2) *Mean Trajectory Length*, the average number of decision steps across all episodes.

### B.5 POLICIES IN EMBODIED TASKS

There are three types of policies in paper: the base policy, the proposal policy, and the revision policy. The base policy is an independent policy that interacts with the environment without using a world model, and when a world model is enabled, it is always the same as the corresponding proposal policy. When a world model is integrated, the proposal policy generates multiple candidate action sequences at each decision step, and the revision policy evaluates these candidates and selects one based on the predicted rollouts produced by the world model.

#### B.5.1 BASE POLICIES AND PROPOSAL POLICIES

In our experiments, we employ two types of base policies for AR and ImageNav: a VLM policy and a heuristic policy. For the VLM policy, we use Qwen2.5-VL-72B-Instruct-AWQ (Bai et al., 2025) as the default base policy and as the proposal policy when integrated with a world model to answer queries. For the heuristic policy, we implement a primitive action sampling mechanism that draws actions from the action space according to the previously executed actions and a set of handcrafted rules. Concretely, if there exists a previous action, then the next action must not be its inverse (for example, a `turn_left` cannot be immediately followed by a `turn_right`). In addition, we prevent excessively long subsequences of turns in the same direction by capping the maximum number of consecutive turns to four. These rules help the heuristic policy to avoid redundant back-and-forth movements and to explore the environment effectively.

For manipulation tasks, we likewise consider two base policies: a VLM policy and a 3D diffusion policy. The VLM policy remains Qwen2.5-VL-72B-Instruct-AWQ by default. The 3D diffusion policy follows 3D Diffuser Actor (Ke et al., 2024); we train it using the authors' official code. To encourage diverse action trajectory proposals, we drop its text input and modify the task-definition scripts so that task variants occur with equal frequency during training. For each manipulation task, the diffusion policy is trained on 120 demonstrations and used as the proposal policy to generate short-horizon 7-DoF gripper action sequences within the planning loop. When using 3D Diffuser Actor as the proposal policy, we report results on only three manipulation tasks, since we find that Stack Cups is difficult for the diffusion policy to learn reliably.

#### B.5.2 REVISION POLICIES

The revision policy is the component that refines the proposals produced by the proposal policy using the world model rollouts. At each decision step $t$, the proposal policy outputs $M$ candidate action sequences $\{\hat{\mathbf{A}}_t^{(m)}\}_{m=1}^M$, and the world model predicts the corresponding future observations $\{\hat{\mathbf{O}}_t^{(m)}\}_{m=1}^M$. The revision policy

$$\pi_{\text{revision}} : \left( \{(\hat{\mathbf{A}}_t^{(m)}, \hat{\mathbf{O}}_t^{(m)})\}_{m=1}^M, \mathbf{o}_t, \text{g} \right) \mapsto \mathbf{D}_t^\star$$

consumes these imagined trajectories together with the current observation $\mathbf{o}_t$ and goal g, and outputs the final decision $\mathbf{D}_t^\star$. Depending on the task, $\mathbf{D}_t^\star$ may be a pure action decision (ImageNav, Manipulation) or a joint action–answer decision (AR, A-EQA).

**Score-and-select for action-only tasks.** For Image-Goal Navigation and Robotic Manipulation, the objective is to reach a goal state, and the revision policy only needs to choose which action sequence to execute. In these settings, $\mathbf{D}_t^\star = \hat{\mathbf{A}}_t^\star$ and $\pi_{\text{revision}}$ is instantiated as a score-and-select operator as in Equation (4) of the main paper:

$$\hat{\mathbf{A}}_t^\star = \hat{\mathbf{A}}_t^{(m^\star)}, \quad m^\star = \arg \max_{m \in \{1, \dots, M\}} S_{\text{act}}\left( \hat{\mathbf{A}}_t^{(m)}, \hat{\mathbf{O}}_t^{(m)} \,\middle|\, \mathbf{o}_t, \text{g} \right),$$

where $S_{\text{act}}(\cdot)$ is an action-centric scoring function that estimates the expected task reward of each imagined trajectory (e.g., progress toward the goal).

In most experiments, we instantiate $S_{\text{act}}$ with a VLM-based reward model: we use Qwen2.5-VL-72B-Instruct-AWQ as the default revision policy to score candidate rollouts and to select the action sequence with the highest predicted utility. For ablations, we also replace Qwen2.5-VL-72B-Instruct-AWQ with InternVL3-78B-AWQ (Zhu et al., 2025); results in Table 7 show that world model integration consistently improves performance regardless of the specific VLM used. In addition to

Table 7: Task performance for InternVL3 variants with and without a world model. Higher **SR%**, **SPL%**, and **Ans. Score** are better; lower **Mean Traj.** is better.

| Model Details | | AR | | ImageNav | | | A-EQA | | |
|---|---|---|---|---|---|---|---|---|---|
| Model Type | Method | SR $\uparrow$ | Mean Traj. $\downarrow$ | SR $\uparrow$ | Mean Traj. $\downarrow$ | SPL $\uparrow$ | Ans. Score $\uparrow$ | Mean Traj. $\downarrow$ | SPL $\uparrow$ |
| Base Policy | InternVL3 (w/o WM) | 49.91 | 7.06 | 13.19 | 60.30 | 7.46 | 47.28 | 20.45 | 31.22 |
| + Image Gen. | SVD† | 55.72 | 5.37 | 40.97 | 52.50 | 26.26 | 47.13 | 16.78 | 34.54 |

VLM-based scoring, we consider task-specific reward functions when a direct signal is available. For example, in Image-Goal Navigation we also evaluate an LPIPS-based reward that measures perceptual distance between predicted observations and the goal image, and use this score in place of the VLM-based $S_{\text{act}}$.

**Joint action–answer refinement for AR and A-EQA.** For AR and A-EQA, each episode combines action planning and question answering. Here, the world model rollouts are used not only to guide the next action, but also to provide auxiliary visual evidence for the final answer (e.g., multi-view observations that reduce occlusions). This leads to a richer instantiation of the revision policy than the pure score-and-select operator above.

At time step $t$, the output of $\pi_{\text{revision}}$ is decomposed into an action component and an answer component. Let $\hat{y}_t$ denote the predicted answer (a category label for AR and a natural-language answer for A-EQA). We write

$$\mathbf{D}_t^{\star} = \left(\hat{\mathbf{A}}_t^{\star}, \hat{y}_t\right) = \pi_{\text{revision}}\Big(\{(\hat{\mathbf{A}}_t^{(m)}, \hat{\mathbf{O}}_t^{(m)})\}_{m=1}^M, \mathbf{o}_t, \mathbf{g}\Big).$$

In our implementation, the action component $\hat{\mathbf{A}}_t^{\star}$ is still selected by a score-and-select rule with an action scoring function $S_{\text{act}}$:

$$\hat{\mathbf{A}}_t^{\star} = \hat{\mathbf{A}}_t^{(m^{\star})}, \quad m^{\star} = \underset{m \in \{1, \ldots, M\}}{\arg\max} \, S_{\text{act}}\Big(\hat{\mathbf{A}}_t^{(m)}, \hat{\mathbf{O}}_t^{(m)} \,\big|\, \mathbf{o}_t, \mathbf{g}\Big),$$

while the answer component $\hat{y}_t$ is obtained by aggregating predicted futures from all candidates:

$$\hat{y}_t = f_{\text{ans}}\Big(\mathbf{o}_t, \mathbf{g}, \{\hat{\mathbf{O}}_t^{(m)}\}_{m=1}^M\Big).$$

Here, $S_{\text{act}}(\cdot)$ again scores trajectories from the perspective of future task performance (for example, preferring trajectories that move the agent toward informative views or closer to the target object), and $f_{\text{ans}}(\cdot)$ is an answer head that consumes the current observation, the goal, and the set of predicted futures as multi-view evidence. In practice, $f_{\text{ans}}$ is implemented with the same vision-language model as the proposal policy, which takes the frames as input and outputs the answer.

Thus, for AR and A-EQA, the revision policy operates in two coupled ways: it chooses how the agent should move next via $S_{\text{act}}$ and $\hat{\mathbf{A}}_t^{\star}$, and it simultaneously uses the simulated rollouts as additional context to produce a more informed answer $\hat{y}_t$. This joint action–answer refinement is a richer instantiation of $\pi_{\text{revision}}$ than the score-only operator in Equation (4), and is specific to tasks that require both control and question answering.

## B.6 WORLD MODELS IN EMBODIED TASKS

**Output format.** The world models evaluated in our framework fall into two categories according to their native output format: *perspective* models and *panoramic* models. Perspective models, such as NWM (Bar et al., 2025), LTX-Video (HaCohen et al., 2024), and Wan2.1 (Wan et al., 2025), generate frames in a perspective view. Panoramic models, including PathDreamer (Koh et al., 2021), SE3DS (Koh et al., 2023), and our post-trained variants, produce equirectangular panoramas. For integration into our closed-loop pipeline, panoramic outputs are decomposed into perspective views, which are then supplied to the agent. In A-EQA, the agent consumes four principal perspective views (front, left, right, back) when they are available. In AR, the agent uses the view that contains the target bounding box; if the box is not visible, we discard the generated frames until the predicted box (from SAM2) enters the field of view. Unless otherwise specified, each perspective view image is resized to $384 \times 384$ pixels before being passed to the agent.

**Input format.** Panoramic models are conditioned on an equirectangular panorama at a resolution of $576 \times 1024$ pixels. Perspective models, when possible, take the current front-view observation with resolution $480 \times 480$ as input. Some models require additional modalities. SE3DS expects a depth map, while PathDreamer requires both depth and a per-pixel semantic label map. For all depth-aware models, we provide ground-truth depth from Habitat. For PathDreamer, the initial semantic map is obtained by running a pretrained RedNet (Jiang et al., 2018) on the initial RGB-D frame to produce per-pixel labels that match the required input specification.

## C   POST-TRAINING RECIPE FOR EMBODIED WORLD MODELS

In this section, we describe how an off-the-shelf video generation model is adapted, via post-training, into an action-controllable world model suitable for embodied tasks. We first formalize the learning objective and the action-observation alignment (Appendix C.1), and then detail the concrete post-training setup used for tasks in Habitat-Sim and for Robotic Manipulations (Appendix C.2).

### C.1   PROBLEM FORMULATION

Let $\mathbf{x}_1 \in \mathbb{R}^{3 \times H \times W}$ denotes the initial RGB frame that conditions the generation process. Our goal is to synthesize an $N$-frame video $\mathbf{X} = [\mathbf{x}_1, \mathbf{x}_2, \ldots, \mathbf{x}_N] \in \mathbb{R}^{3 \times H \times W \times N}$, where $\mathbf{X}$ represents a plausible sequence of future observations after executing a sequence of actions $\mathbf{A} = [a_1, a_2, \ldots, a_N]$.

For tasks in Habitat-Sim, we adopt a discrete action space with $a_i \in \mathcal{V}$, where $\mathcal{V}$ is a finite set of navigation primitives (e.g., `Forward`, `Turn-Left`, `Turn-Right`, `Stop`). For manipulation, we use a continuous action space with $a_i \in \mathbb{R}^7$, corresponding to 7-DoF end-effector poses. Actions in Habitat-Sim specify relative transformations between consecutive observations. Since $a_i$ maps $\mathbf{x}_{i-1}$ to $\mathbf{x}_i$, no action precedes the first frame. To maintain a one-to-one alignment between frames and actions, we prepend a special token and set $a_1 = a_{\text{Null}}$. In contrast, for manipulation tasks during post-training, actions are absolute end-effector poses expressed in the world frame, so there is naturally a one-to-one correspondence between actions and frames.

We formulate future-observation synthesis with the world model $g_{\boldsymbol{\theta}}$ by learning the conditional distribution $p_{\boldsymbol{\theta}}(\mathbf{X} \,|\, \mathbf{x}_1, C(\mathbf{A}))$, where $C(\mathbf{A})$ denotes the control signal emitted by the unified action API. This API converts the native action sequence $\mathbf{A}$ into the conditioning interface expected by the pretrained video generator (for example, a text prompt, a camera trajectory, or a sequence of low-level controls). This formulation yields action-conditioned rollouts that evolve from the initial frame $\mathbf{x}_1$ according to the specified action sequence, thereby aligning the pretrained model with the domain distribution and action space of the target embodied tasks.

### C.2   POST-TRAINING CONFIGURATION

Table 8: Post-trained (action-conditioned) world models used in our experiments, with repositories and training configurations.

| World Model | Domain | Repository | Frames ($N$) | Train Res. | Notes |
|---|---|---|---|---|---|
| **Post-training on Habitat-Sim data** | | | | | |
| Cosmos-Predict2† (Agarwal et al., 2025) | Habitat-Sim | github.com/nvidia-cosmos/cosmos-predict2 | 13 | $576 \times 1024$ | Official repo |
| LTX-Video† (HaCohen et al., 2024) | Habitat-Sim | github.com/Lightricks/LTX-Video-Trainer | 17 | $576 \times 1024$ | Official repo |
| Wan2.1† (Wan et al., 2025) | Habitat-Sim | github.com/modelscope/DiffSynth-Studio | 13 | $576 \times 1024$ | Official repo |
| Wan2.2 (5B)† (Wan et al., 2025) | Habitat-Sim | github.com/modelscope/DiffSynth-Studio | 13 | $576 \times 1024$ | Official repo |
| Wan2.2 (A14B)† (Wan et al., 2025) | Habitat-Sim | github.com/modelscope/DiffSynth-Studio | 13 | $576 \times 1024$ | Official repo |
| SVD† (Blattmann et al., 2023a) | Habitat-Sim | github.com/pixeli99/SVD_Xtend | 14 | $576 \times 1024$ | Self-adapted based on repo |
| **Post-training on manipulation data** | | | | | |
| Cosmos-Predict2† (Agarwal et al., 2025) | Manipulation | github.com/nvidia-cosmos/cosmos-predict2 | 13 | $480 \times 480$ | Official repo |
| SVD† (Blattmann et al., 2023a) | Manipulation | github.com/pixeli99/SVD_Xtend | 14 | $448 \times 448$ | Self-adapted based on repo |

For tasks in Habitat-Sim, we use panoramic observations as both the input and the output of the video generators. We fine-tune the pretrained video generation models at a resolution of $576 \times 1024$ and train them to predict $N$ future frames on our self-collected panoramic action-observation corpus from Habitat-Sim. In these tasks, the action space is discrete and comprises four navigation primitives: `Forward 0.2m`, `Turn_Left 22.5°`, `Turn_Right 22.5°`, and `Stop`. For manipulation tasks, we use front-view observations as both the input and the output of the video generators. We fine-tune the pretrained video generation models at a resolution of $480 \times 480$ (Cosmos-Predict2) or

$448 \times 448$ (SVD) and train them to predict $N$ future frames with continuous 7-DoF end-effector poses as conditioning.

Unless otherwise stated, post-training uses 40K sampled instances for the Habitat-Sim tasks and for the manipulation tasks. All models are initialized from their official pretrained weights and adapted on the corresponding dataset for one epoch. We rely on the official implementations and the recommended hyperparameters for fine-tuning whenever available; specific post-training details of various world models are summarized below in Tables 8 and 9.

Table 9: All the world models and their details in World-In-World. "†" denotes post-trained (action-conditioned) variants.

| World Model | Model Type | Control Type | Input Type | #Param. |
|---|---|---|---|---|
| **Zero-shot (no post-training)** | | | | |
| PathDreamer (Koh et al., 2021) | Image Gen. | Viewpoint | RGB-D; Pano | 0.69B |
| SE3DS (Koh et al., 2023) | Image Gen. | Viewpoint | RGB-D; Pano | 1.1B |
| NWM (Bar et al., 2025) | Video Gen. | Trajectory | RGB | 1B |
| SVD (Blattmann et al., 2023a) | Video Gen. | Image | RGB | 1.5B |
| LTX-Video (HaCohen et al., 2024) | Video Gen. | Text | RGB | 2B |
| Hunyuan (Kong et al., 2024) | Video Gen. | Text | RGB | 13B |
| Wan2.1 (Wan et al., 2025) | Video Gen. | Text | RGB | 14B |
| Wan2.2 (Wan et al., 2025) | Video Gen. | Text | RGB | 5B |
| Wan2.2 (Wan et al., 2025) | Video Gen. | Text | RGB | A14B |
| Cosmos-Predict2 (Agarwal et al., 2025) | Video Gen. | Text | RGB | 2B |
| Runway Gen4 (Runway Research, 2025) | Video Gen. | Text | RGB | – |
| **Post-trained (action-conditioned)** | | | | |
| SVD† (Blattmann et al., 2023a) | Video Gen. | Action | RGB; Pano | 1.5B |
| LTX-Video† (HaCohen et al., 2024) | Video Gen. | Action | RGB; Pano | 2B |
| Wan2.1† (Wan et al., 2025) | Video Gen. | Action | RGB; Pano | 14B |
| Wan2.2† (Wan et al., 2025) | Video Gen. | Action | RGB; Pano | 5B |
| Wan2.2† (Wan et al., 2025) | Video Gen. | Action | RGB; Pano | A14B |
| Cosmos-Predict2† (Agarwal et al., 2025) | Video Gen. | Action | RGB; Pano | 2B |

In Table 10, we summarize the computational resources required to post-train each world model on ∼40k domain-specific clips collected from Habitat-Sim. This post-training stage is intentionally lightweight and is several orders of magnitude less expensive than full pretraining. For 14B-parameter variants, we adopt LoRA fine-tuning to reduce GPU memory usage, while all other models are fine-tuned with full weights.

Table 10: Post-training resources for ∼40k domain clips per model. The procedure is lightweight and substantially cheaper than full retraining.

| Model | Model Size | GPU Memory (peak) | H100 GPU-hours |
|---|---|---|---|
| SVD | 1.5B | 84 GB | 29 |
| LTX-Video | 2B | 61 GB | 5 |
| Wan2.1 | 14B | 57 GB | 74 |
| Cosmos-Predict2 | 2B | 71 GB | 15 |

## D  POST-TRAINING DATASET CONSTRUCTION

For the post-training dataset used in manipulation tasks, we rely on the official RLBench code-base (James et al., 2020) to generate data. Specifically, we produce 200 demonstrations for each manipulation task. Each demonstration includes approximately 150 front-view RGB observations together with the corresponding sequence of 7-DoF end-effector poses. These pose sequences are aligned with the image observations and serve as the action labels during post-training. For the tasks evaluated in Habitat-Sim (Savva et al., 2019), there is no existing pipeline for constructing a large-scale dataset of panoramic action trajectories. To address this gap, we build a comprehensive post-training dataset by sampling action trajectories from the training splits of indoor scenes in HM3D (Ramakrishnan et al., 2021) and Matterport3D (Chang et al., 2017). Our trajectory sampling procedure is described in Appendix D.1. A summary of the resulting dataset statistics is provided in Table 11.

### D.1  TRAJECTORY SAMPLING

---

**Algorithm 1** Three-stage construction of the post-training panoramic dataset

---

**Input:** scene mesh $\mathcal{S}$, waypoint density $\rho$, weight $\alpha$, filter radius $r_f$, leaf ratio $\eta$
**Output:** set of panoramic trajectories $\mathcal{T}$

    // Stage 1: waypoint selection
1:  $S \leftarrow \text{Area}(\mathcal{S})$
2:  $N_{\text{wp}} \leftarrow \max\big(1400, \lfloor \rho S \rfloor\big)$                                          ▷ target number of points
3:  $\mathcal{P} \leftarrow \textsc{UniformSampleNavigable}(\mathcal{S}, N_{\text{wp}})$
4:  build geodesic distance matrix $D$ on $\mathcal{P}$
5:  **for all** $p_i \in \mathcal{P}$ **do**                                       ▷ leaf score $s(i)$
6:     $\text{ecc}(i) \leftarrow \max_j D_{ij}$
7:     $\bar{d}(i) \leftarrow \frac{1}{|\mathcal{P}|-1} \sum_j D_{ij}$
8:     $s(i) \leftarrow \text{ecc}(i) + \alpha\, \bar{d}(i)$
9:  sort $\mathcal{P}$ by $s(i)$ in descending order               ▷ higher $s(i)$ = more peripheral
10: $\mathcal{W} \leftarrow \varnothing$
11: **for all** $p_i$ in sorted $\mathcal{P}$ **do**                    ▷ radius-based greedy pruning
12:     **if** $\forall w \in \mathcal{W} : D_{iw} \geq r_f$ **then**
13:         $\mathcal{W} \leftarrow \mathcal{W} \cup \{p_i\}$
    // Stage 2: path generation
14: $\mathcal{T} \leftarrow \varnothing$
15: $N_{\text{leaf}} \leftarrow \lceil \eta N_{\text{wp}} \rceil$
16: $\mathcal{U} \leftarrow \mathcal{W}[:N_{\text{leaf}}]$                              ▷ unvisited waypoints
17: $c \leftarrow \textsc{RandomSample}(\mathcal{U})$                       ▷ random start
18: **while** $\mathcal{U} \neq \varnothing$ **do**
19:     $n \leftarrow \arg\min_{w \in \mathcal{U} \setminus \{c\}} \textsc{GeodesicDist}(c, w)$
20:     $\tau \leftarrow \textsc{ShortestPath}(c, n)$                  ▷ Habitat planner
21:     record panoramic RGB-D frames along $\tau$ and append to $\mathcal{T}$
      // Stage 3: waypoint dynamic update
22:     **for all** $w \in \mathcal{W}$ **do**
23:         **if** $\exists m \in \tau : \textsc{GeodesicDist}(m, w) < r_f$ **then**
24:             $\mathcal{W} \leftarrow \mathcal{W} \setminus \{w\}$                ▷ mark as visited
25:     recompute $s(\cdot)$ on updated $\mathcal{W}$, then sort in descending order
26:     $\mathcal{U} \leftarrow \mathcal{W}[:N_{\text{leaf}}]$                    ▷ refresh unvisited set
27:     $c \leftarrow n$
28: **return** $\mathcal{T}$

---

Our aim is to record physically reasonable trajectories that resemble the exploration behavior of real agents in indoor spaces. We follow three guiding principles: (i) *Diversity*. The trajectories should cover many viewpoints and actions so that the model sees the scene from different perspectives and motion patterns. (ii) *Plausibility*. The paths must respect physical constraints; the agent must not move through walls or other solid objects. (iii) *Manageability*. The data should be free of excessive redundancy so that training remains balanced and efficient.

We implement these principles with a sampling procedure shown in Algorithm 1 and described below.

| Statistic | Value |
|---|---|
| Number of scenes | 858 |
| Panorama RGB frames | 763,724 |
| Action trajectories | 439,213 |
| Depth recorded | ✓ |
| Camera poses recorded | ✓ |
| Low-level actions recorded | ✓ |

Table 11: Statistics of the post-training panoramic dataset.

1. **Waypoint selection.** For a scene of floor area $S$ we set the waypoint density to $\rho = 4\,\text{m}^{-2}$ and draw

$$N_{\text{wp}} = \max\big(1400, \lfloor \rho S \rfloor\big)$$

navigable points $\mathcal{P}$ uniformly across the scene. We construct a complete graph whose edge weights $D_{ij}$ are the geodesic distances between points $p_i$ and $p_j$. Each vertex $i$ is assigned a leaf score

$$s(i) = \text{ecc}(i) + \alpha\, \bar{d}(i),$$

where $\mathrm{ecc}(i) = \max_j D_{ij}$ is the eccentricity, $\bar{d}(i) = (|\mathcal{P}| - 1)^{-1} \sum_j D_{ij}$ is the mean geodesic distance to all other vertices, and $\alpha = 1.7$. Sorting vertices by $s(i)$ in descending order, we greedily build a waypoint set $\mathcal{W}$ that respects a minimum spacing of $r_{\mathrm{f}} = 3\,\mathrm{m}$: a candidate $v$ is accepted only if $D_{vj} \geq r_{\mathrm{f}}$ for every waypoint $j$ already chosen.

2. **Path generation.** We maintain a list $\mathcal{U}$ of unvisited waypoints, initialized with the top $N_{\mathrm{leaf}}$ vertices of $\mathcal{W}$. Starting from a random waypoint $c \in \mathcal{U}$, we repeatedly move to the nearest unvisited waypoint

$$n = \arg \min_{w \in \mathcal{U} \setminus \{c\}} \mathrm{GEODESICDIST}(c, w),$$

and use the Habitat path-finder to compute the shortest collision-free path $\tau$ from $c$ to $n$. Panoramic RGB-D frames are recorded at every step along $\tau$ and appended to the trajectory set $\mathcal{T}$.

3. **Waypoint dynamic update.** After each segment $\tau$ we label any waypoint $w$ with $\mathrm{GEODESICDIST}(m, w) < r_{\mathrm{f}}$ for some path point $m \in \tau$ as *visited* and remove it from $\mathcal{W}$. We then recompute $s(\cdot)$ on the remaining vertices, resort $\mathcal{W}$, and refresh the unvisited list

$$\mathcal{U} \leftarrow \mathcal{W}[:N_{\mathrm{leaf}}].$$

The next segment starts from $c \leftarrow n$, and the loop continues until $\mathcal{U}$ is empty. This dynamic reselection guarantees that peripheral regions are covered while avoiding redundant sampling in interior corridors.

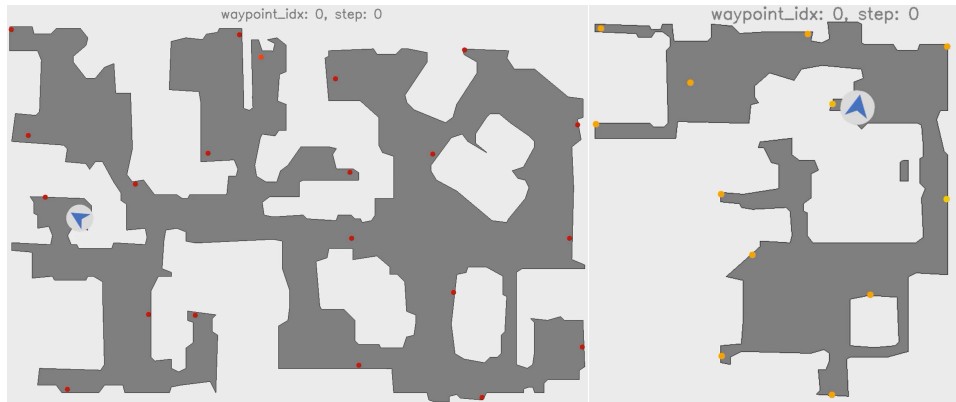

Figure 12: Top-down visualization of sampled waypoints in a scene. Red (left) and yellow (right) dots are the final waypoints after radius-based pruning. The proposed strategy places waypoints throughout peripheral regions while avoiding redundant interior points, yielding diverse and spatially balanced trajectories.

Compared with random sampling of start and end waypoints, the above strategy distributes waypoints across peripheral areas such as bedrooms while avoiding redundant paths through interior corridors. The resulting dataset therefore offers a balanced and diverse set of viewpoints for post-training (see Figure 12).

## E    VISUALIZING WORLD MODEL PREDICTIONS

We illustrate the behavior of several world models under identical action sequences generated by the planner. Figure 13 and Figure 14 show example rollouts in which the action sequence consists solely of Forward actions; a well-behaved model should yield pure forward motion. The figures contrast models that follow the commands with those that drift or hallucinate, underscoring the importance of precise action control for downstream embodied tasks. These examples also reveal current limitations of world models in trustworthy prediction (Sun et al., 2023; Zhang et al., 2025c; Mei et al., 2025). For further examples of good and bad predictions, see Figures 15 to 18.

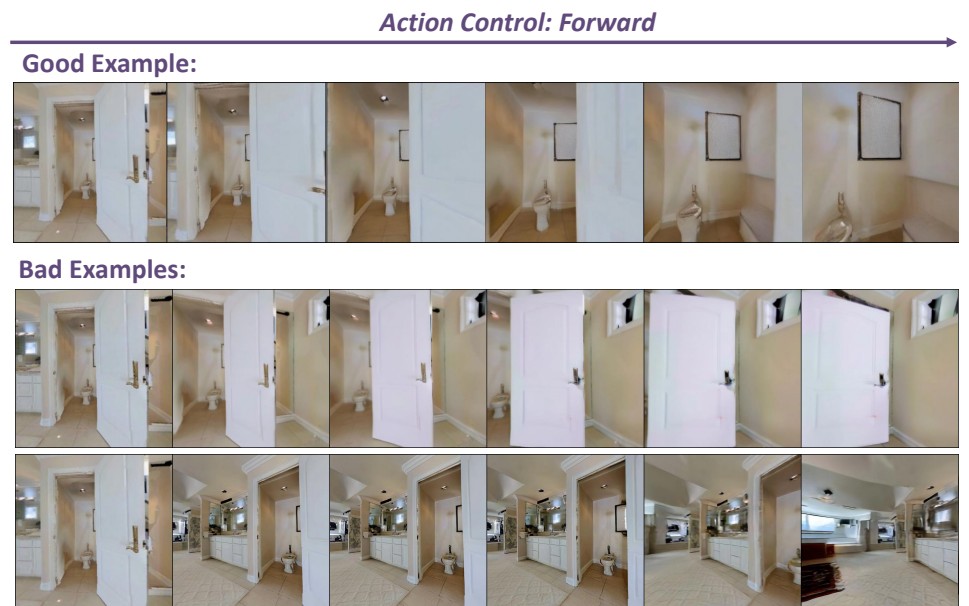

Figure 13: Examples of good and bad predictions. The action sequence contains only `Forward` actions. Models that violate this requirement yield observations that can mislead the planner.

Figure 14: Examples of good and bad predictions. The action sequence contains only `Forward` actions. Models that violate this requirement yield observations that can mislead the planner.

**Good Examples:**

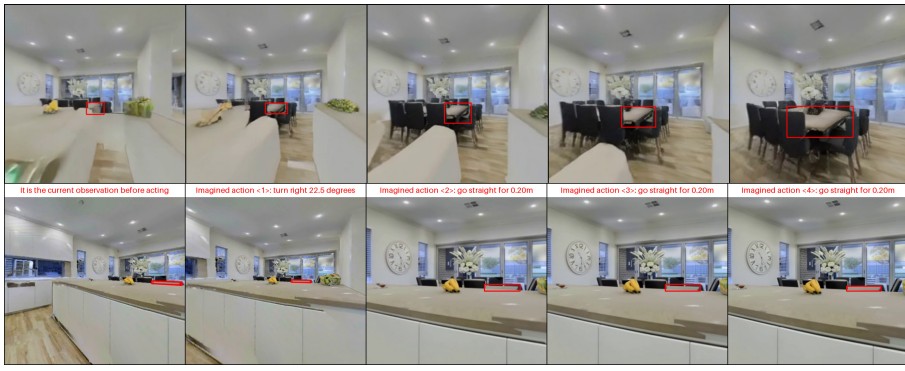

**Bad Examples:**

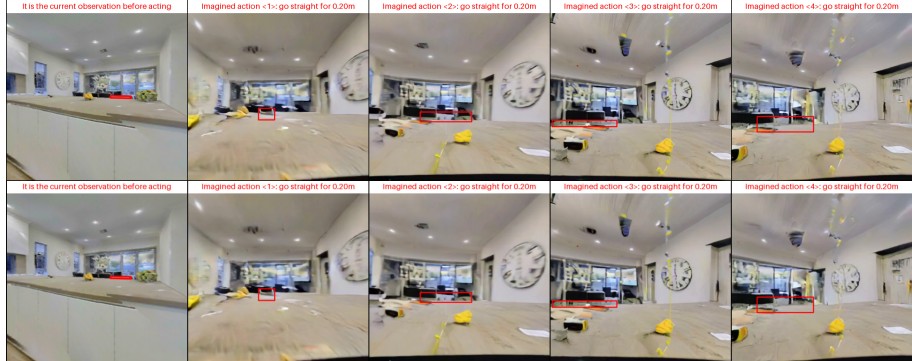

Figure 15: Additional examples of good and bad predictions.

**Good Examples:**

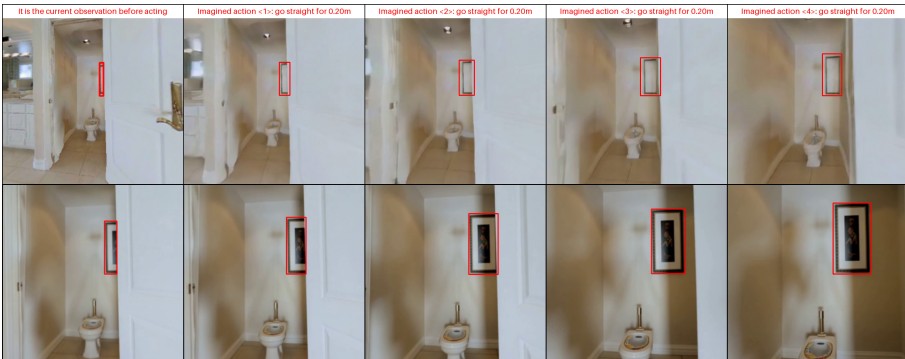

**Bad Examples:**

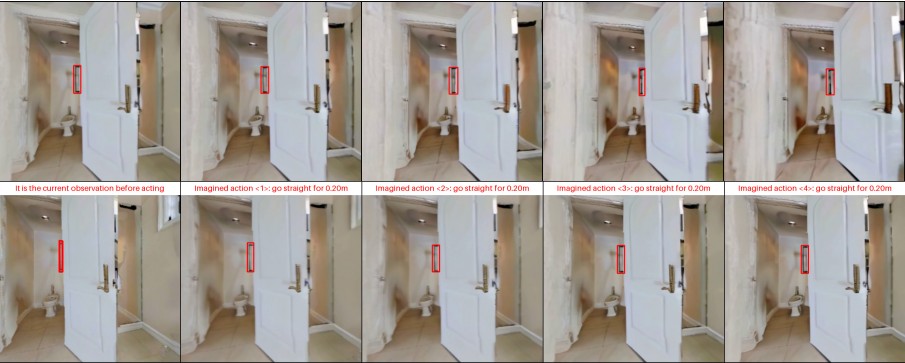

Figure 16: Additional examples of good and bad predictions.

**Good Examples:**

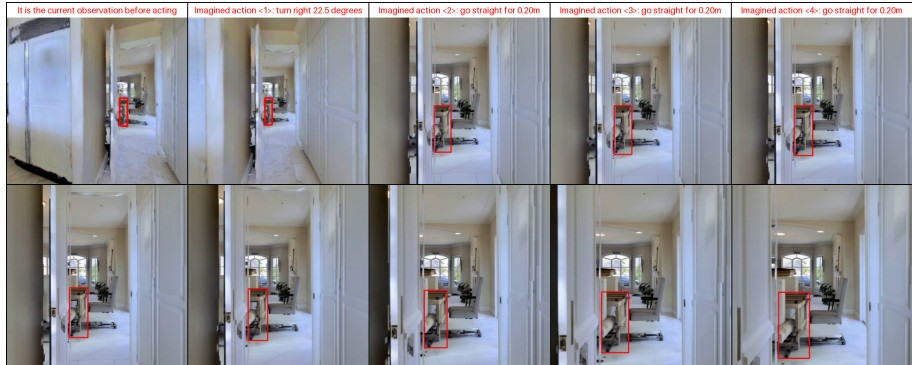

**Bad Examples:**

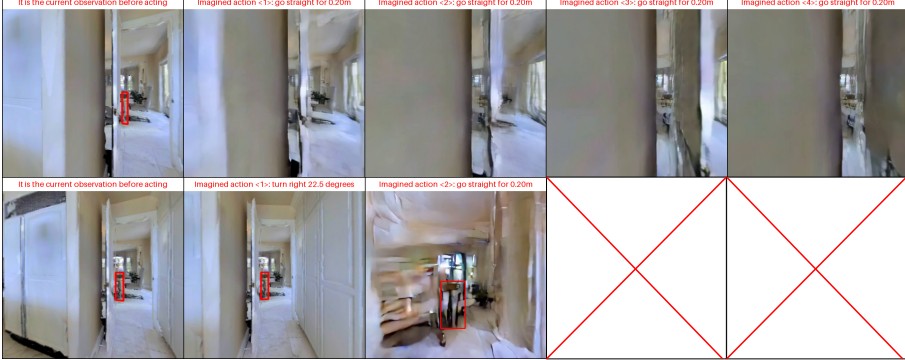

Figure 17: Additional examples of good and bad predictions.

**Good Examples:**

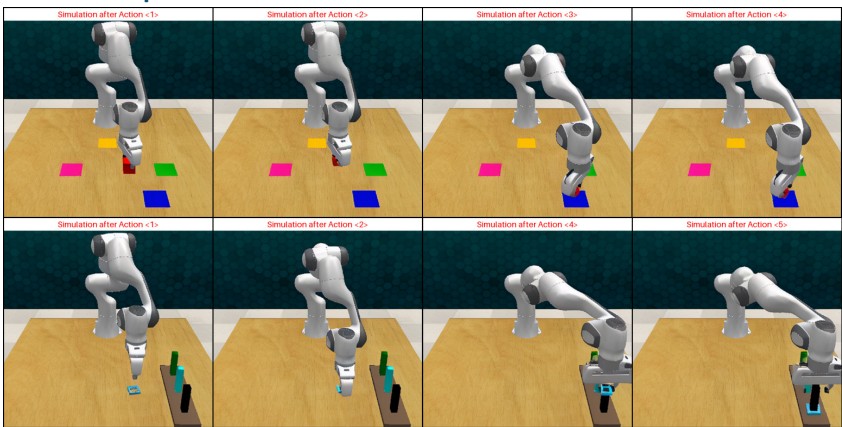

**Bad Examples:**

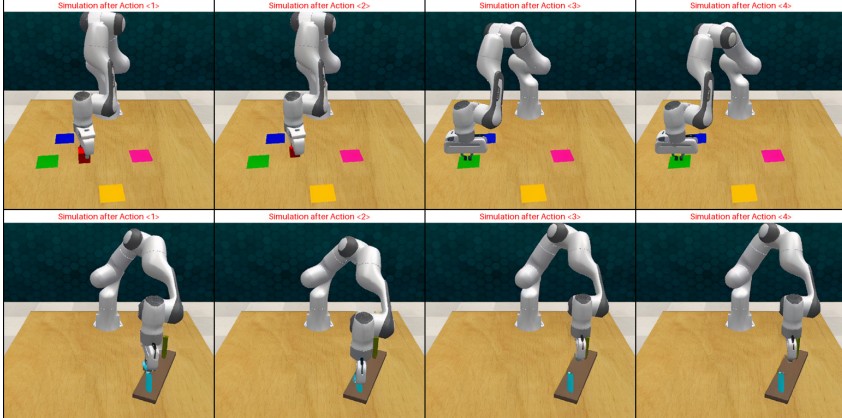

Figure 18: Additional examples of good and bad predictions.

## F    PROMPT TEMPLATES USED IN WORLD-IN-WORLD

In this section, we provide the exact prompt templates used in our experiments for four tasks in World-In-World: (i) Active Recognition (AR), (ii) Image-Goal Navigation (ImageNav), (iii) Active Embedded Question Answering (A-EQA), and (iv) Robotic Manipulation.

### F.1    ACTIVE RECOGNITION (AR) PROMPT

---

**AR Answerer Prompt**

Please recognize the object in the image bounded by the red box.

---

**AR Planner Prompt**

You are an AI agent tasked with identifying a target object within an image—specifically, the object enclosed by a red bounding box.
Your objective is to navigate toward a viewpoint that maximizes the target's visibility and recognition accuracy.
**Instructions:**

1. Based on the current `{obs_key}` observation, plan the next `<look_ahead_action_num>` action(s) to take in sequence.

2. Use the following heuristics to guide planning:

    - If the red-boxed object appears on the left side of the image, turning left often improves visibility.
    - If it appears on the right side, turning right is usually beneficial.
    - If the object is partially occluded or obstructed, consider repositioning to bypass the obstacle and refine your viewpoint.

3. Choose a sequence of actions that leads to a clear, centered, and unobstructed view of the red-boxed object.

---

**AR Answerer Additional Prompt (with WM Rollouts)**

You now have a composite visualization formed by stitching imagined views from multiple perspectives around your current position. These perspectives are centered on the target object (enclosed within the red bounding box).
Use these synthesized views to:

- Improve object identification accuracy.
- Make more informed recognition decisions.

---

**AR Planner Additional Prompt (with WM Rollouts)**

You are now simulating imagined future trajectories by generating hypothetical actions and their corresponding observations.
Use these imagined observations to:

- Evaluate the potential outcomes of different action sequences.
- Make informed navigation decisions by selecting the next best action based on predicted future states and your current state.

**Note:**

- Each imagined frame is annotated with the specific action taken and its index at the top of the image.
- Pay attention to the presence of red bounding boxes indicating the target object. If the target is not visible in a frame, this indicates a poor action.

---

> • You should adjust your action selection strategy to avoid such failure states.

## F.2 IMAGE-GOAL NAVIGATION (IMAGENAV) PROMPT

---

**ImageNav Planner Prompt**

You are an AI navigation agent tasked with locating the position from which the **goal image** was captured. Your objective is to plan a sequence of actions that leads to a position where the goal image is clearly visible, centered in the front view, and appears to have been taken within at your current position.
**Inputs:** You are provided with a sequence of images:

1. First, the current egocentric observation: `{obs_key}`.

2. Last, the **goal image**: a reference image that represents the target viewpoint you are trying to reach.

**Task:**

1. Based on the input images, plan the next `<{look_ahead_action_num}>` action(s) in order.

2. Optimize for:

   - **Alignment**: The goal image should be centered in the front view.
   - **Proximity**: Your position should match the goal image's capture point.
   - **Visibility**: The goal image should appear clear and unobstructed in your current front view.

---

**ImageNav Planner Prompt (with WM Rollouts)**

You are an AI navigation agent tasked with locating the position from which the **goal image** was captured. Your objective is to plan a sequence of actions that leads to a position where the goal image is clearly visible, centered in the front view, and appears to have been taken within at your current position.
**Inputs:** You are provided with:

1. The **goal image**: a reference image that represents the target viewpoint you are trying to reach.

**Task:**

1. Based on the input images, plan the next `<{look_ahead_action_num}>` action(s) in order.

2. Optimize for:

   - **Alignment**: The goal image should be centered in the front view.
   - **Proximity**: Your position should match the goal image's capture point.
   - **Visibility**: The goal image should appear clear and unobstructed in your current front view.

---

## F.3 ACTIVE EMBEDDED QUESTION ANSWERING (A-EQA) PROMPT

---

**A-EQA High-Level Planner Prompt**

You are an embodied navigation and question-answering agent specialized in indoor scene understanding. Your goal is to either answer the user's question directly from the current observation or propose a high-level navigation planning to gather more information.
**User Query:**
`{question}`

---

**Inputs:**
You are provided with the following:

1. A **stitched panoramic image with annotations** — composed of multiple directional images captured from your current position (the name of each view is labeled on the top of the image). Each detected object is annotated with its contour and a unique object index.

2. A **stitched panoramic image without annotations** — visually identical but without overlays, serving as a clean reference.

3. A dictionary mapping detected objects to their corresponding perspective views and object indices in the annotated image:
   *Format:* `{{view_id:  {{object_index:  object_name}}}}`
   *Current mapping:* **{detected_objs}**

Note all the provided images are in the formt of `{obs_key}`.

**Task Description:**
Your task is to:

1. Analyze the visual information from each perspective direction.

2. Identify all possible **exits** and **doorways** in the environment.

3. Give one high-level **navigation plan** to further explore the scene in order to answer the User Query.

4. If the answer to the question is fully evident from the current observation, provide it directly. Otherwise, set your current answer to "None".

**Output Format:**
Return your response as a dictionary with the following structure:     {
`'Reason':  <Your visual reasoning and analysis>,`
`'Action Plan':  <Description of your next high-level`
`navigation plan>,`
`'Chosen View':  <One of:  'front', 'left', 'right', or`
`'back', indicating the view you are going to further explore`
`in your Action Plan>,`
`'Chosen Landmark':  <Index of the selected object landmark`
`from the annotated stitched image, or 'None'>`
`'Answer':  <Your answer to the User Query, or 'None'>`
`}`

**Constraints:**

- Provide **exactly one** high-level action, including one `'Chosen View'` and one `'Chosen Landmark'`.

- If no suitable annotated object is available in your desired direction, set `'Chosen Landmark'` to `'None'` and describe your intended action in the `'Action Plan'` field.

- Each `'Action Plan'` should include a **clear and executable instruction and stop condition**.
  – Good Example: `'Action Plan':  "Pass through the doorway (object index "3") in the front view, and stop once inside the next room."`
  – Good Example: `'Action Plan':  "Approach the sofa (object idx "10") in the left view, and stop once we can see the objects on it."`
  – Bad Example: `'Action Plan':  "Move into the kitchen area visible in the view and stop once inside the kitchen."` – kitchen area is not a specific object and not clear how to get there.

- If a landmark is selected, it must correspond to a **visible, annotated object** in the stitched image.

- Do not select unlabeled objects — they typically indicate previously visited or non-informative regions.

- Populate 'Answer' **only** when you are confident the question can be answered from the current observation. Otherwise, set 'Answer': 'None' in the dictionary.

**Tips:**

- If you observe a door in a closed state, it means you cannot pass through it.

- If the current observation shows that your previous plan has not yet been completed, it is acceptable to propose a similar plan again to continue pursuing the same goal.

- Leverage human spatial habits to guide your planning. For instance, if the goal involves finding a television, selecting a nearby sofa may be effective, as these often appear together in living spaces.

---

**A-EQA Low-Level Planner Prompt**

You are now performing **low-level navigation action planning** for an indoor scene exploration task.

**Inputs:**

You are provided with:

1. An updated **RGB image with annotations**, representing the egocentric view of your current environment:
   - Detected objects are annotated with contours and unique object indices with square text boxes.

2. A **high-level navigation plan** represented as a dictionary with two fields:
   - 'Action Plan': A description of the intended navigation strategy.
   - 'Chosen Landmark': The object index of the selected landmark from the annotated image to approach, or 'None' if no landmark is selected (in which case follow the 'Action Plan' description).

   The current high-level plan is: **{high_level_plan}**

Note all the provided images are in the format of {obs_key}.

**Task:**

Your task is to:

1. Analyze the visual scene and identify your position relative to the goal.

2. Determine the next **low-level action(s)** to take in sequence, up to a maximum of **<{look_ahead_action_num}>** steps.

**Constraints:**

- You must generate **less than {look_ahead_action_num}** low-level actions.

- The actions sequence should align with the goal described in **high-level 'Action Plan' and 'Chosen Landmark'**.

- If the navigation goal or selected landmark in the high-level plan is either:
  - not visible in the current observation, or
  - already reached (i.e., centered, unobstructed, and close),

  then your only action should be "stop".

**Tips:**

- If the landmark object is partially occluded or obstructed, consider repositioning to bypass the obstacle before approaching it directly.

- Choose actions that meaningfully move the agent toward the selected landmark or fulfill the intent of the high-level plan.

- Maintain spatial awareness: understand the relationship between your egocentric view and the direction of the target.

---

**A-EQA High-Level Planner Additional Prompt (with WM Rollouts)**

In addition to your current (real) observations, you are now provided with **simulated outcomes**—low-resolution reconstructions that represent the potential result of executing future navigation plans. These simulated outcomes are designed to help you better understand your surroundings and support more informed navigation planning.

**Each simulated outcome includes:**

- **Proposed High-Level Plan**: A hypothetical navigation strategy used to generate the simulated result.

- **Simulated Observation**: A stitched panoramic image showing what the environment might look like after following the proposed plan.

You should use this information to:

- Evaluate the potential effectiveness and correctness of the proposed high-level strategies.

- Make informed decisions by selecting your next high-level plan based on both the **simulated information** and your **current real observation**.

**Notes:**

- Object indices remain consistent across simulated and real observations.

- Simulated outcomes are **NOT fully accurate**. If you believe you can answer the user query based on simulation alone, you should **NOT** provide a final answer yet. Instead, select a high-level plan that will lead to a real observation and validate your answer afterward.

Your current **simulated outcomes** are:

---

## F.4 ROBOTIC MANIPULATION PROMPT

---

**Manipulation Planner Prompt**

You are a Franka Panda robot with a parallel gripper. You can perform various tasks and output a sequence of gripper actions to accomplish a given task with images of your status. The input space, output action space and color space are defined as follows:

**Input Space**

You are given the following inputs:

1. **Human Instruction**: A natural language command specifying the manipulation task goal.

2. **Object Dictionary**:
   - Each object is represented by a unique index (e.g., object 1) and mapped to a 3D discrete coordinate [X, Y, Z].

3. **Annotated Scene Image**:
   - Each object in the image is annotated with:
     - A circle point marker with
     - A unique object index, which corresponds to the object dictionary.
   - There is a red XYZ coordinate frame located in the **top-left corner** of the table.
     - The **XY plane** represents the surface plane of the table (Z = 0).
     - The valid coordinate range for X, Y, Z is: [0, {}].

**Output Action Space**

- Each output action is represented as a 7D discrete gripper action in the following format: `[X, Y, Z, Roll, Pitch, Yaw, Gripper state]`.

- X, Y, Z are the 3D discrete position of the gripper in the environment. It follows the same coordinate system as the input object coordinates.

---

- The allowed range of X, Y, Z is [0, {}].
- Roll, Pitch, Yaw are the 3D discrete orientation of the gripper in the environment, represented as discrete Euler Angles.
- The allowed range of Roll, Pitch, Yaw is [0, {}] and each unit represents {} degrees.
- Gripper state is 0 for close and 1 for open.

**Color space**

- Each object can only be described using one of the colors below:
  ```
  ["red", "maroon", "lime", "green", "blue", "navy",
  "yellow", "cyan", "magenta", "silver", "gray", "olive",
  "purple", "teal", "azure", "violet", "rose", "black",
  "white"],
  ```

{}

---

**Manipulation Planner Additional Prompt (with WM Rollouts)**

You are now provided with **simulated outcomes** in addition to your real-time observations. These outcomes are low-resolution predictions of what the scene may look like after executing hypothetical action plans.

They are intended to help you reason about the environment and make more informed decisions.

**Simulated Outcome Structure**

Each simulated-outcome item includes:

- **Proposed Action Plan**: The sequence of gripper actions that led to the simulated result.
- **Simulated Observation**: The simulated result after following the proposed plan.

**How to Use This Information**

You must consider both:

1. Your current **real observation** of the environment, and
2. The provided **simulated outcomes**.

Use these to:

- Evaluate how well each proposed plan satisfies the task objective.
- Identify if any proposed plan fully achieves the instruction goal.
- If a proposed plan appears valid and effective, **you may adopt it directly** as your final response.
- If no plan fully meets the goal, **generate a revised or entirely new action plan**, guided by insights from the simulations and the real-world scene.

**Additional Notes**

- Simulated outcomes are **approximate**. Treat them as helpful forecasts, not absolute truth.
- You must analyze these hypothetical action plans and their simulated outcomes in the `reasoning_and_reflection` field of the returned JSON (e.g., their differences and why you choose one over another).
- Always prioritize correctness and robustness in the final executable plan.

You are now given the following **simulated outcomes**:

## G   Use of Language Models

We used large language models strictly as writing assistants for language refinement: grammar correction, style tightening, phrasing alternatives, and minor reorganization for clarity and brevity. No prompts involved technical ideation, modeling, implementation, data analysis, or result selection. All suggested edits were reviewed by the authors, and the technical content, experiments, results, and conclusions are author-generated and author-validated. LLM assistance did not affect the substance of the work.

