# OpenReview forum: "World-In-World: World Models in a Closed-Loop World"
_ICLR.cc/2026/Conference — ICLR 2026 Oral_

### Official Review · Reviewer_Bvsj · 2025-10-23

**Soundness:** 3
**Presentation:** 3
**Contribution:** 3
**Rating:** 8
**Confidence:** 4

**Summary:**

This paper introduces WoW!, the first open benchmark for evaluating generative world models in a closed-loop, control-focused setting across four domains: Active Recognition, Image-Goal Navigation, Active Embodied Question Answering, and Robotic Manipulation. The authors propose a unified planning strategy and action API to integrate diverse world models into a consistent decision-making pipeline. Key findings include: (1) controllability is more crucial than visual quality for task success; (2) post-training with limited data yields significant gains; and (3) increasing inference-time compute improves performance. This paper shifts the evaluation focus from visual fidelity to embodied decision-making, setting a new standard for world model assessment in interactive environments.

**Strengths:**

- This paper is well-written and easy to follow.
- It addresses a crucial gap in the evaluation of video generation models from a control-centric perspective, an area previously overlooked by benchmarks, which is a valuable step forward in world model development.
- The benchmark’s task design and baseline selection are thorough and rigorous, incorporating both high-level and low-level decision-making tasks, as well as the latest video generation models.
- The experimental insights are impactful, especially the finding that controllability is more important than visual quality for decision-making tasks, and the observation that model-based planning is scalable during test-time.

**Weaknesses:**

- The analysis of proposal and revision policies is limited. Specifically, the revision policy, which selects among simulated rollouts, uses the same VLM as the proposal policy. The paper does not explore how sensitive the results are to the revision policy design, such as whether a simpler reward model would yield different outcomes for world model utility.
- The unified action API assumes world models can conform to a shared action vocabulary. However, real-world domains often use incompatible action semantics, and mapping between them may introduce misalignment or information loss. The paper does not analyze how such action misalignments affect final task performance.
- Another concern is the computational cost and inference-time overhead of the proposed framework. Planning-based methods are known for being computationally intensive, and integrating large video generation models into decision-making could significantly reduce the frequency of decision-making due to high computational demands.

**Questions:**

- How much impact do the revision policies have on task success?
- How sensitive is closed-loop performance to action misalignment caused by the unified action API? Could the different action API lead to different experimental conclusions?
- Compared to the base policy, how much does the inference time increase when introducing world models? How can the computational burden of model-based planning be mitigated?

---

> ### Author Response · Authors · 2025-11-19
> **Response to Reviewer Bvsj [1/3]**
>
> We thank the reviewer’s feedback and would provide further clarification.
>
> > **W1, Q1**: The analysis of revision policies is limited. The revision policy uses the same VLM as the proposal policy, and the paper does not explore sensitivity to revision policy design, such as using a simpler reward model.
>
> **TL;DR: We add experiments with a simple LPIPS-based revision policy for Image-Goal Navigation and find that WMs still provide large gains over the VLM-only baseline, and in some cases even outperform the VLM-based revision policy. This shows that our conclusions are not tied to using a powerful VLM as the revision policy.**
>
> Thank you for the valuable comment. We agree that analyzing the impact of different revision policy designs on task success is important. In response, we conducted additional experiments that use a simple LPIPS-based reward function as the revision policy for Image-Goal Navigation, instead of the universal VLM-based revision policy. Concretely, during planning, after obtaining the $M$ action plans and their corresponding imagined observations from the proposal policy and WMs, we compute the average LPIPS between the predicted observations and the goal image for each imagined trajectory. We then select the action plan with the lowest LPIPS distance as the final action to execute. This is similar in spirit to NWM [6], which also uses LPIPS as a reward function to score imagined trajectories for image-goal navigation, but here we implement and evaluate it within our unified framework and closed-loop setting. The results are as follows:
>
> | Proposal Policy | WM Augmentation      | Revision Policy | ImageNav SR↑ | ImageNav Mean Traj.↓ | ImageNav SPL↑ |
> | --------------- | -------------------- | --------------- | ------------ | -------------------- | ------------- |
> | VLM             | None (w/o WM)        | None            | 35.42        | 47.5                 | 25.88         |
> | VLM             | SVD†     | VLM             | 43.05        | 46.0                 | 30.96         |
> | VLM             | WAN2.1†  | VLM             | 45.14        | 45.8                 | 32.10         |
> | VLM             | SVD†     | LPIPS           | 47.92        | 41.3                 | 39.82         |
> | VLM             | WAN2.1†  | LPIPS           | 48.61        | 39.8                 | 42.48         |
>
> From these results, we observe that even with a simple LPIPS-based revision policy, incorporating WMs still yields significant improvements over the VLM-only baseline without WM augmentation. For example, using WAN2.1† with the LPIPS-based revision policy achieves an ImageNav success rate of 48.61%, compared to 35.42% for the VLM-only baseline, and it also outperforms the universal VLM-based revision policy (45.14%). This supports our main claim that the predictive capabilities of WMs provide consistent benefits for decision-making, independent of the specific revision policy design.
>
> Moreover, the LPIPS-based revision policy is computationally efficient, since it avoids additional VLM queries and relies on a deterministic perceptual similarity measure. This suggests that when simpler and deterministic reward models are available, our framework can still effectively leverage WM predictions for planning on image-goal navigation tasks. We have included these new results and the corresponding analysis in the revised manuscript.

---

> ### Author Response · Authors · 2025-11-19
> **Response to Reviewer Bvsj [2/3]**
>
> > **W2, Q2**: The unified action API assumes world models can conform to a shared action vocabulary. However, real-world domains often use incompatible action semantics, and mapping between them may introduce misalignment or information loss. The paper does not analyze how such action misalignments affect final task performance.
>
> **TL;DR: We agree that action misalignment is inevitable in a unified interface, but we (i) design mappings to be one-to-one whenever possible, (ii) tune text prompts for text-controlled WMs, and (iii) empirically show that performance variation from prompt-induced misalignment is modest, while WM gains remain significant.**
>
> Thank you for the valuable comment. We agree that action misalignment introduced by the unified action API can affect the evaluation of WMs, especially when different WMs expose heterogeneous control interfaces. Our goal is to provide a fair comparison under a shared action space while minimizing information loss when mapping to each WM’s native interface.
>
> For text-controlled WMs, mapping discrete actions to text commands can indeed introduce mismatch. The resulting text sequences may differ from those seen during pretraining and may express the same semantic action with slightly different phrasing, which can lead to suboptimal behavior. However, such misalignment is largely unavoidable when comparing WMs that operate over different control modalities in a unified framework. To mitigate this, we carefully design the mapping from the unified action space to each WM’s action interface. Concretely:
> - For text-based WMs, we test multiple prompt templates that verbalize the same discrete action sequence and select the best-performing one on validation environments.
> - For WMs that accept structured actions as input, we make the mapping one-to-one so that there is no information loss. For example, in Habitat-Sim tasks, the policy outputs predefined discrete actions (e.g., `turn_left` by 22.5°, `move_forward` by 0.2 m). For WMs that accept discrete actions, we adopt exactly the same action space for post-training and evaluation, so that the WM input actions are identical to the policy actions (e.g., the WM also receives `turn_left` by 22.5°, `move_forward` by 0.2 m). These design choices aim to reduce the negative impact of action misalignment and ensure that heterogeneous WMs are evaluated as fairly as possible within our framework.
>
> To further quantify the impact of action misalignment for text-based interfaces, we report results on the AR task using different text prompt templates for WMs that take text commands as input:
>
> | Text Prompt Template                                    | Wan2.1, AR SR↑ | Cosmos-P2, AR SR↑ |
> | ------------------------------------------------------- | -------------- | ----------------- |
> | "Follow this sequence of camera motions: {action_seq}." | 58.26          | 55.35             |
> | "Follow the actions to move: {action_seq}."             | 57.79          | 55.17             |
>
> These results show that while prompt wording does have a measurable effect, the variation in success rate is relatively modest across templates.

---

> ### Author Response · Authors · 2025-11-19
> **Response to Reviewer Bvsj [3/3]**
>
> > **W3, Q3**: Another concern is the computational cost and inference-time overhead of the proposed framework. Compared to the base policy, how much does the inference time increase when introducing world models? How can the computational burden of model-based planning be mitigated?
>
> **TL;DR: We disscuss inference-time overhead for different WMs and find that lightweight models like LTX-Video introduce moderate overhead, while large models like Wan2.1 incur higher costs. We discuss several strategies to mitigate computational burden, including efficient WM inference, adaptive planning, efficient architectures, and distillation.**
>
> We agree that understanding the computational cost of different WMs is important for practical deployment. In our framework, the inference-time overhead depends on both the choice of WM and the planning hyperparameters (e.g., number of rollouts and planning horizon). In practice, we observe that lightweight WMs such as LTX-Video introduce only a moderate overhead (on the order of 20–30% compared to the base policy), whereas large video generators such as Wan2.1 can increase inference time by a factor of 2–3×. This trade-off is expected: stronger generative models provide higher-fidelity rollouts but are more expensive per step.
>
> We see our work as a first step toward understanding the benefits of WM-based planning; reducing its computational cost is an important direction for future work. We list several potential strategies to mitigate the burden:
>
> (1) **More efficient WM inference.** Recent “self-forcing” style methods [1,2] propose autoregressive video diffusion with reduced computation compared to full-sequence diffusion, which must process all tokens for every denoising step. Incorporating such efficient inference schemes into WMs would directly reduce the per-rollout cost.
>
> (2) **Adaptive planning mechanisms.** Instead of using a fixed number of rollouts and a fixed planning depth, the agent can adaptively decide when and how much to plan based on task difficulty or model uncertainty. For simple states, the policy may act directly without WM rollouts, while for ambiguous or high-stakes states, it can invoke deeper planning. This is analogous to “thinking” in LLMs, where the model uses chain-of-thought only when needed to solve more complex queries.
>
> (3) **More efficient WM architectures.** Designing WMs that explicitly balance speed and quality can further reduce overhead. For example, recent efficient video generation architectures such as StreamDit [3] aim at real-time or streaming generation with lower computational cost, and similar design principles can be applied to WMs used for planning.
>
> (4) **Distillation to smaller WMs.** Distillation techniques can compress large WMs into smaller models that retain the predictive capabilities while being much faster at inference time [4,5]. Such distilled WMs could be plugged into our framework as drop-in replacements to provide more practical planning modules.
>
> We also have added a discussion of the observed overhead and these mitigation strategies to the revised manuscript.
>
> ---
>
> ## References:
>
> [1] Xun Huang, Zhengqi Li, Guande He, Mingyuan Zhou, and Eli Shechtman. Self forcing: Bridging the train-test gap in autoregressive video diffusion. ArXiv, 2506.08009, 2025.
>
> [2] Justin Cui, Jie Wu, Ming Li, Tao Yang, Xiaojie Li, Rui Wang, Andrew Bai, Yuanhao Ban, and Cho-Jui Hsieh. Self-forcing++: Towards minute-scale high-quality video generation. ArXiv, 2510.02283, 2025.
>
> [3] Akio Kodaira, Tingbo Hou, Ji Hou, Masayoshi Tomizuka, and Yue Zhao. Streamdit: Real-time streaming text-to-video generation. ArXiv, 2507.03745, 2025.
>
> [4] Niket Agarwal, Arslan Ali, Maciej Bala, Yogesh Balaji, Erik Barker, Tiffany Cai, Prithvijit Chattopadhyay, Yongxin Chen, Yin Cui, Yifan Ding, et al. Cosmos world foundation model platform for physical AI. arXiv preprint arXiv:2501.03575, 2025.
>
> [5] Hanyang Wang, Fangfu Liu, Jiawei Chi, and Yueqi Duan. Videoscene: Distilling video diffusion model to generate 3d scenes in one step. arXiv preprint arXiv:2504.01956, 2025.
>
> [6] Amir Bar, Gaoyue Zhou, Danny Tran, Trevor Darrell, and Yann LeCun. Navigation world models. In Proceedings of the IEEE/CVF Conference on Computer Vision and Pattern Recognition (CVPR), 2025.

---

> > ### Comment · Reviewer_Bvsj · 2025-11-23
> >
> > Thank you for your detailed reply. Most of my concerns have been addressed. However, I still suggest that the authors show the detailed inference computational cost in the main text or the appendix. I keep my score as 8.

---

> > > ### Author Response · Authors · 2025-11-24
> > > **Response to Bvsj**
> > >
> > > Thank you for your continued engagement and constructive feedback. We appreciate your suggestion to include detailed inference computational costs for the different WMs an now we provide this analysis below and will include it in the Appendix of the revised manuscript.
> > >
> > > We measured the inference time of each WM on a fixed hardware setup (single NVIDIA H100 GPU) and defined a simple **cost-performance** metric as
> > > $
> > > \text{cost-performance} = \frac{\Delta \text{AR SR}}{\text{Inference time}},
> > > $
> > > where $\Delta \text{AR SR}$ is the improvement in AR task success rate over the VLM-only baseline (without WM augmentation). For fairness, we fix the number of rollouts per step to 3 for all WMs and report the corresponding inference time and gains:
> > >
> > > | Model name        | Resolution   | Inference time | Δ AR SR↑ | cost performance↑ |
> > > |-------------------|-------------:|---------------:|---------:|------------------:|
> > > | SVD†              | 1024×576     | 16 s           | +10.71   | 0.67              |
> > > | Wan2.1† 14B       | 1024×576     | 65 s           | +12.34   | 0.19              |
> > > | LTX-Video†        | 1024×576     | 7 s            | +7.26    | 1.04              |
> > > | Wan2.2† A14B      | 1024×576     | 65 s           | +12.16   | 0.19              |
> > > | Cosmos-P2†        | 1024×576     | 17 s           | +9.98    | 0.59              |
> > > | Wan2.2 A14B       | 480×480      | 83 s           | +9.26    | 0.11              |
> > > | Hunyuan           | 480×480      | 17 s           | +7.44    | 0.44              |
> > > | NWM               | 224×224      | 38 s           | +7.08    | 0.19              |
> > > | LTX-Video         | 480×480      | 3 s            | +5.81    | 1.94              |
> > > | Wan2.1            | 480×480      | 53 s           | +7.99    | 0.15              |
> > > | Cosmos-P2         | 480×480      | 12 s           | +5.08    | 0.42              |
> > > | SVD               | 480×480      | 10 s           | +7.44    | 0.74              |
> > >
> > > From this table, we see that LTX-Video and LTX-Video† achieve the highest cost-performance among all WMs, meaning they provide the largest AR success-rate improvement per unit of inference time. This suggests that lightweight WMs with efficient architectures can be very attractive in settings with tight compute or latency budgets.
> > >
> > > In contrast, larger models such as Wan2.1 and Wan2.2 deliver strong absolute gains in SR, but their much longer inference times reduce their cost-performance, which may limit their practicality in real-time or resource-constrained deployments. In comparsion, lightweight, fast architectures like LTX-Video offer a promising direction for “Lightweight World Models” that can still provide meaningful planning benefits under strict compute budgets.
> > >
> > > We sincerely appreciate your time and thoughtful evaluation. Thank you again.

---

### Official Review · Reviewer_KANk · 2025-10-27

**Soundness:** 3
**Presentation:** 3
**Contribution:** 4
**Rating:** 8
**Confidence:** 4

**Summary:**

This work addresses a significant gap in evaluating world models for their utility in the development of embodied agents tasks. World models are largely evaluated for their visual quality, motion coherence or controllability, but there is no available benchmark to assess how well a world model can improve embodied reasoning and task success. This work introduces a new general evaluation benchmark for world models, specific for assessing their performance on embodied interactions. The authors provide 1) a unified closed-loop planning strategy 2) a unified action API to allow various world models to be compared and 3) a study on finetuning pretrained video generators for a specific downstream task and the effect of data scaling.

**Strengths:**

1) Addresses an existing gap in the research community on assessing world models for their ability to be integrated in decision making embodied agents, presenting a well-rounded general framework for closed-loop evaluation. This allows the research community to make informed decisions when pursuing research goals around improving the utility of world models for embodied scenarios.
2) Presents a comprehensive set of tasks for embodied agents - 4 tasks (active recognition, image-goal navigation, active embodied question answering, robotic manipulation) - I particularly appreciate the inclusion of robotic manipulation, as it provides an important additional dimension and usability for the proposed evaluation framework.
3) Conducts an extensive and detailed analysis on 9 world model benchmarks, both using image and video based approaches, highlighting the key results of their evaluation: the use of world models enhance provides performance gains over base policies across tasks, post training aids the utility of world models & the current limitations in manipulation tasks where world models struggle with simulating precise motion and dynamics.
4) Presents valuable insights on data-size scaling for post-training, highlighting that even a small dataset size (4K) from the task domain can yield significant gains with post training. This is useful for future work in this space.

**Weaknesses:**

Overall, the work provides a great set of references and supporting experiments. The paper is easy to follow, with detailed result figures and tables.

A couple of minor suggestions:
- I might have missed this, but I couldn't see the results of the A-EQA task evaluation from Table 2 discussed in more detail in Section 3. There is a stronger focus on the other 3 tasks, specifically AR, ImageNav and robotic manipulation.
- In Fig 5(b), Wan2.2 5B (post-trained) seems to have high controllability, but performs low in terms of task success rate, which goes against the positive correlation claimed in the paper. Same for Runway Gen4 that has lower LIPIS, but very high SR.
- At the beginning of section 3.2, you discuss findings presented in Figure 5 (a) - it would be good to reference it.

**Questions:**

1) In Table 2, what does A14B stand for in the Wan2.2 #param column?
2) nitpick: For clarity, it might be good to mark in the main body text on line 303 that post-trained variants have a special symbol (described in the description of Table 1), and first used on line 313 with Wan2.1.

**Details Of Ethics Concerns:**

No ethics concerns

---

> ### Author Response · Authors · 2025-11-19
> **Response to Reviewer KANk [1/1]**
>
> We appreciate the reviewer’s constructive feedback and would like to provide additional clarification.
>
> > **W1, W3**: A-EQA task evaluation results from Table 2 are not discussed in detail in Section 3. At the beginning of section 3.2, should also reference Figure 5(a) when discussing findings.
>
> We agree that our discussion of A-EQA in Section 3 was relatively brief compared to other tasks, and that not referencing Figure 5(a) at the beginning of Section 3.2 made the narrative less clear. In the uploaded revised manuscript, we have expanded the analysis of the A-EQA results in Sections 3.2 and 3.3, and we now explicitly reference Figure 5(a) when discussing the main findings.
>
> > **W2**: In Fig 5(b), Wan2.2 5B (post-trained) has high controllability but low task success rate, which contradicts the claimed positive correlation. Same for Runway Gen4 that has lower LPIPS but very high SR.
>
> **TL;DR: We provide further analysis of these two models, Runway Gen4 and Wan2.2 5B†, to explain why they deviate from the overall positive trend between controllability and task success rate.**
>
> We appreciate the reviewer for highlighting these cases. We agree that while there is an overall positive **trend** between controllability (measured by 1-LPIPS) and task success rate (SR), individual models such as Runway Gen4 and Wan2.2 5B† deviate from this trend. We now provide further analysis about them:
>
> - **Runway Gen4.**
>   Runway Gen4 is a strong proprietary model. In our embodied tasks, even when used in a zero-shot setting without post-training, it achieves the highest SR among all models, including post-trained open-source WMs. However, its reconstruction metric (1-LPIPS) is not the best. Based on visual inspection of rollouts, we speculate that Gen4 has been trained on large, high-quality synthetic video datasets, especially indoor scenes. When evaluated on our real-world reconstructed indoor environments (HM3D / Matterport3D), its generated videos have a noticeable style difference compared to ground-truth observations. This style gap can hurt pixel-level reconstruction metrics like LPIPS, but the rollouts remain semantically meaningful and contextually aligned with the underlying scene and task. As a result, Gen4 can support strong decision-making and achieve high SR despite its lower reconstruction score. We show qualitative examples for AR in our interactive demo (linked from the project page), where this effect is visible.
>
> - **Wan2.2 5B†.**
>   Wan2.2 5B† illustrates the opposite situation. Its controllability metric (1-LPIPS) is high, but its SR is lower than one might expect from that score. In general, more physically faithful and controllable rollouts tend to help SR. However, task success also depends on whether the WM captures a **useful distribution of plausible futures**, not just similarity to a single ground-truth outcome. Under partial observability, the most helpful prediction is often one that is semantically aligned with ground truth, even if it is not pixel-close to the ground-truth frame.
>
>   For example, consider a chair partly occluded by a table. With limited views, a strong WM may infer a reasonable chair pose and appearance from context and priors (clearly “chair-like” and consistent with the scene), while a weaker WM may generate a patchy texture or unrealistic shape. Both may receive similar pixel-wise scores because neither matches the exact ground-truth view, but only the former supports robust recognition, navigation, and control. Our interpretation is that Wan2.2 5B† tends to produce rollouts that are less aligned with the semantics and affordances needed for decision-making than some other models. This leads to relatively lower SR despite a high controllability score, which is the mirror case of Runway Gen4.
>
> These two examples highlight that while 1-LPIPS is a useful proxy for controllability, it is not a perfect predictor of downstream task success. We will clarify this nuance in the revised manuscript and emphasize that our main claim is about a general positive **trend**, with notable exceptions that further motivate the need for closed-loop evaluation.
>
> > **Q1, Q2**: In Table 2, what does A14B stand for in the Wan2.2 #param column? For clarity, it might be good to mark in the main body text on line 303 that post-trained variants have a special symbol.
>
> Thank you for pointing this out. “A14B” is the official parameter type used by the Wan2.2 model provider, and we follow this naming convention in our table. The “A14B” label indicates a mixture-of-experts configuration with two 14B-parameter experts, while the active parameter size at inference time is effectively 14B.
>
> In the revised manuscript, we (i) add a short explanation of “A14B” in the Table 2 caption and the model description, and (ii) explicitly mark post-trained variants with the † symbol in both the table and the main text to make the distinction between base and post-trained models clear.

---

### Official Review · Reviewer_8Zz4 · 2025-10-29

**Soundness:** 3
**Presentation:** 4
**Contribution:** 3
**Rating:** 6
**Confidence:** 3

**Summary:**

This article presents the Wow! World Models in a Closed-Loop World, which is the first Visual World Model (WM) evaluation platform for Closed-Loop interactive scenarios. Unlike previous open-loop evaluations that focused solely on the quality of the generated video —such as video fidelity —Wow, taking task success rate as the core index, the practical utility of the world model for decision-making is systematically evaluated across four embodied intelligent tasks (active recognition, image target navigation, active question answering, and manipulator operation). The authors introduce a unified online planning strategy and a standardized action API to enable heterogeneous world models to participate in closed-loop interactions under the same protocol, three key findings were revealed: (1) visual quality ≠ task success, and controllability is more important; (2) fine-tuning with a small amount of action-observation data on the target domain is more effective than upgrading the pre-trained video generator; (3) increasing the number of simulations can significantly improve the performance.

**Strengths:**

- The author has pointed out the “Open-loop bias” in the current world model evaluation,i.e., overemphasis on generation quality and neglect of its practical value in closed-loop decision-making. Wow! For the first time, the focus of assessment has shifted from “Looking like” to “Working with,” a paradigm shift that plays an important guiding role for embodied AI and the world model community.
- This paper constructs four tasks covering perception, navigation, and operation, and evaluates more than ten mainstream world models (including SVD, Wan, LTX, Cosmos, etc.). Through the standardized action API and online planning framework, heterogeneous models can be fairly compared.
- The conclusion that “High visual quality ≠ high task success” challenges the current community's prevailing assumptions about generative model capabilities (such as whether Sora-like models are naturally suitable for embodied tasks). The authors further provide empirical support through the controllability quantification (LPIPS alignment), the data size law, and the inference extension, and the results show that the proposed model is more robust than previous models. It points to the optimization direction for follow-up research (e.g., emphasizing alignment of motion conditions rather than simply improving resolution).

**Weaknesses:**

- In the current framework, both the proposal and revision policies use powerful VLM (such as QWEN2.5-VL-72B). This makes the extent to which task success is attributable to the world model vs. The strategy itself is unclear. Complementary ablation experiments are suggested: for example, fixed-strategy capabilities (such as using weak or regular strategies), observing differences in performance gains across different world models, to more purely assess the decision-aid value of WM.
- Habitat-sim data comes only from indoor scenes in HM3D and Matterport3D, and RLBench is also limited to desktop operations. This limits generalization to more complex environments, such as outdoor, dynamic obstacles, and multi-agents. The authors mention generalizability as a challenge in the discussion section but do not validate cross-domain transfer in experiments (e.g., in Habitat training and the Gibson test), weakening the persuasiveness of the universal world model claim.

**Questions:**

- In table 1-3, the baseline strategy (e.g., VLM) already has strong capabilities (e.g., AR task 50% + accuracy). Can the world model still deliver significant improvements with weaker strategies (such as random strategies or simple heuristics)? This helps clarify the marginal contribution of WM.
- The paper highlights generalization as a key challenge (Section 4), but all experiments were evaluated within the training domain. Have you tried to train the model after testing in an unfamiliar scenario (e.g., new building layout, new object category) or across simulators (e.g., Habitat → Igibson)? Preliminary results, even if negative, are very useful.
- Figure 7 shows that increasing the number of inferences improves performance, but computing resources are limited in real deployments. What is the “cost performance”(performance/amount of computation) of the different world models on a fixed computing budget (e.g. 5 rollouts per step)? Are there design implications for a Lightweight World Model?

---

> ### Author Response · Authors · 2025-11-19
> **Response to Reviewer 8Zz4 [1/3]**
>
> We thank the reviewer’s feedback and would provide further clarification.
>
> > **W1, Q1**: Both proposal and revision policies use powerful VLMs (QWEN2.5-VL-72B), making it unclear how much task success comes from the world model versus the policy itself. Ablation experiments with weaker baseline policies would better isolate WM contributions.
>
> **TL;DR: We have conducted additional experiments using simpler heuristic proposal policies and LPIPS-based revision policies to better isolate the contribution of WMs. The results show that WMs provide significant benefits regardless of the specific policy design.**
>
> *1. Different Proposal Policies*
>
> We agree this is an important concern. To better isolate the contribution of the WM from that of the policy, we have already conducted experiments with simpler heuristic policies as baselines in our initial submision. Specifically, we implemented rule-based proposal policies for AR and ImageNav, and evaluated their performance with and without WM augmentation. For clarity, we reorganize the relevant results from the main paper as follows:
>
> | Proposal Policy         | WM Augmentation      | AR SR↑ | AR Mean Traj.↓ | ImageNav SR↑ | ImageNav Mean Traj.↓ | ImageNav SPL↑ |
> | ------------------- | -------------------- | ------ | -------------- | ------------ | -------------------- | ------------- |
> | Heuristic           | None (w/o WM)        | 39.02  | 8.81           | 2.08         | 59.60                | 0.63          |
> | Heuristic           | SVD†     | 60.62  | 5.17           | 20.83        | 58.50                | 11.86         |
> | Heuristic           | WAN2.1†  | 62.98  | 4.71           | 22.92        | 58.70                | 11.63         |
> | ------------------- | -------------------- | ------ | -------------- | ------------ | -------------------- | ------------- |
> | VLM                 | None (w/o WM)        | 50.27  | 6.24           | 35.42        | 47.50                | 25.88         |
> | VLM                 | SVD†     | 60.98  | 5.02           | 43.05        | 46.00                | 30.96         |
> | VLM                 | WAN2.1†  | 62.61  | 4.73           | 45.14        | 45.80                | 32.10         |
>
> These results show that even when starting from a weak heuristic policy, adding a WM leads to large gains across both AR and ImageNav. For instance, in AR, the heuristic policy’s success rate increases from 39.02% without a WM to 62.98% with WAN2.1†. In ImageNav, the success rate increases from 2.08% to 22.92% under the same WM. While the VLM-based policy achieves higher absolute performance, the **relative** improvements from WM augmentation are substantial for both heuristic and VLM policies.
>
> *2. Different Revision Policies*
>
> In addition, we also conducted additional experiments that use a simple LPIPS-based reward function as the revision policy for Image-Goal Navigation, instead of the universal VLM-based revision policy. Concretely, during planning, after obtaining the $M$ action plans and their corresponding imagined observations from the proposal policy and WMs, we compute the average LPIPS between the predicted observations and the goal image for each imagined trajectory. We then select the action plan with the lowest LPIPS distance as the final action to execute. This is similar in spirit to NWM [1], which also uses LPIPS as a reward function to score imagined trajectories for image-goal navigation, but here we implement and evaluate it within our unified framework and closed-loop setting. The results are as follows:
>
> | Proposal Policy | WM Augmentation      | Revision Policy | ImageNav SR↑ | ImageNav Mean Traj.↓ | ImageNav SPL↑ |
> | --------------- | -------------------- | --------------- | ------------ | -------------------- | ------------- |
> | VLM             | None (w/o WM)        | None            | 35.42        | 47.5                 | 25.88         |
> | VLM             | SVD†     | VLM             | 43.05        | 46.0                 | 30.96         |
> | VLM             | WAN2.1†  | VLM             | 45.14        | 45.8                 | 32.10         |
> | VLM             | SVD†     | LPIPS           | 47.92        | 41.3                 | 39.82         |
> | VLM             | WAN2.1†  | LPIPS           | 48.61        | 39.8                 | 42.48         |
>
> From these results, we observe that even with a simple LPIPS-based revision policy, incorporating WMs still yields significant improvements over the VLM-only baseline without WM augmentation. For example, using WAN2.1† with the LPIPS-based revision policy achieves an ImageNav success rate of 48.61%, compared to 35.42% for the VLM-only baseline, and it also outperforms the universal VLM-based revision policy (45.14%). This supports our main claim that the predictive capabilities of WMs provide consistent benefits for decision-making, independent of the specific revision policy design.

---

> ### Author Response · Authors · 2025-11-19
> **Response to Reviewer 8Zz4 [2/3]**
>
> > **W2, Q2**: Habitat-sim data is limited to scenes in HM3D and Matterport3D, restricting generalization to complex environments. No cross-domain transfer experiments were conducted to validate the universal world model claim.
>
> **TL;DR: We add cross-domain experiments where WMs are post-trained on synthetic HSSD and evaluated on HM3D/Matterport3D. WMs still improve performance under this domain shift, but with a gap to in-domain post-training.**
>
> Thank you for raising this point. We agree that evaluating how well WMs generalize across domains is important.
> To study cross-domain transfer, we have performed additional experiments where we post-train the WMs on data collected from the Habitat Synthetic Scenes Dataset (HSSD) and evaluate them on our existing AR and ImageNav suites based on HM3D and Matterport3D. HSSD is a synthetic dataset with diverse indoor and outdoor assets, and its visual appearance differs significantly from the real-world reconstructed HM3D and Matterport3D scenes. This setting therefore tests whether the post-training phase can learn action-control and visual representations that transfer across a non-trivial domain shift. The results are summarized below:
>
> | Base Policy | WM Augmentation | Post-Training Env. | Test Env.           | AR SR↑ | AR Mean Traj.↓ | ImageNav SR↑ | ImageNav Mean Traj.↓ | ImageNav SPL↑ |
> | ----------- | --------------- | ------------------ | ------------------- | ------ | -------------- | ------------ | -------------------- | ------------- |
> | VLM         | None (w/o WM)   | None               | HM3D/MP3D (val set) | 50.27  | 6.24           | 35.42        | 47.5                | 25.88         |
> | VLM         | SVD†            | HSSD               | HM3D/MP3D (val set) | 58.98  | 5.24           | 38.89        | 47.2                | 27.60         |
> | VLM         | WAN2.1†         | HSSD               | HM3D/MP3D (val set) | 62.98    | 4.78         | 42.36        | 46.0             | 31.18           |
> | VLM         | SVD†            | HM3D (train set)   | HM3D/MP3D (val set) | 60.98  | 5.02           | 43.05        | 46.0                 | 30.96         |
> | VLM         | WAN2.1†         | HM3D (train set)   | HM3D/MP3D (val set) | 62.61  | 4.73          | 45.14         | 45.8                 | 32.10         |
>
> Even though the WMs are post-trained exclusively on synthetic HSSD data, they still provide consistent gains over the VLM-only baseline when evaluated on HM3D and Matterport3D (e.g. For SVD†, AR SR improves from 50.27% to 58.98%, and ImageNav SR improves from 35.42% to 38.89). This indicates that post-training helps WMs acquire more general action-conditioned visual representations that can transfer across domains, in line with observations from Adaworld [2]. At the same time, we do observe some performance drop compared to the setting where post-training and evaluation share the same domain, which is expected under a stronger visual shift.
>
> We will include these cross-domain results in the revised manuscript to better reflect the current empirical scope: our findings show that post-trained WMs can transfer across substantially different scene distributions (HSSD → HM3D/Matterport3D), but there is still a clear gap to fully universal generalization.
>
> Beyond this, we are also collecting data from Gibson scenes and plan to evaluate transfer from Gibson to our HM3D/Matterport3D suites as a follow-up. Gibson is also built from real-world reconstructions (similar to HM3D/MP3D) and may pose an smaller domain gap and performance drop compared to HSSD. So we chose HSSD as our first cross-domain evaluation to better stress-test the generalization ability of WMs.

---

> ### Author Response · Authors · 2025-11-19
> **Response to Reviewer 8Zz4 [3/3]**
>
> > **Q3**: Increasing the number of inferences improves performance, but computing resources are limited in real deployments. What is the “cost performance” (performance/amount of computation) of the different world models on a fixed computing budget (e.g. 5 rollouts per step)? Are there design implications for a Lightweight World Model?
>
> **TL;DR: We measure a simple cost-performance metric (SR gain per second of inference) and find that lightweight models such as LTX-Video and LTX-Video† provide the best performance–compute trade-off, suggesting that efficient architectures are strong candidates for lightweight WMs.**
>
> Thank you for the thoughtful question. We agree that understanding the trade-off between performance and computational cost is crucial for practical deployment of WMs.
>
> To quantify this, we measured the inference time of each WM on a fixed hardware setup (single NVIDIA H100 GPU) and defined a simple **cost-performance** metric as
> $
> \text{cost-performance} = \frac{\Delta \text{AR SR}}{\text{Inference time}},
> $
> where $\Delta \text{AR SR}$ is the improvement in AR task success rate over the VLM-only baseline (without WM augmentation). For fairness, we fix the number of rollouts per step to 3 for all WMs and report the corresponding inference time and gains:
>
> | Model name        | Resolution   | Inference time | Δ AR SR↑ | Cost Performance↑ |
> |-------------------|-------------:|---------------:|---------:|------------------:|
> | SVD†              | 1024×576     | 16 s           | +10.71   | 0.67              |
> | Wan2.1† 14B       | 1024×576     | 65 s           | +12.34   | 0.19              |
> | LTX-Video†        | 1024×576     | 7 s            | +7.26    | 1.04              |
> | Wan2.2† A14B      | 1024×576     | 65 s           | +12.16   | 0.19              |
> | Cosmos-P2†        | 1024×576     | 17 s           | +9.98    | 0.59              |
> | Wan2.2 A14B       | 480×480      | 83 s           | +9.26    | 0.11              |
> | Hunyuan           | 480×480      | 17 s           | +7.44    | 0.44              |
> | NWM               | 224×224      | 38 s           | +7.08    | 0.19              |
> | LTX-Video         | 480×480      | 3 s            | +5.81    | 1.94              |
> | Wan2.1            | 480×480      | 53 s           | +7.99    | 0.15              |
> | Cosmos-P2         | 480×480      | 12 s           | +5.08    | 0.42              |
> | SVD               | 480×480      | 10 s           | +7.44    | 0.74              |
>
> From this table, we see that LTX-Video and LTX-Video† achieve the highest cost-performance among all WMs, meaning they provide the largest AR success-rate improvement per unit of inference time. This suggests that lightweight WMs with efficient architectures can be very attractive in settings with tight compute or latency budgets.
>
> In contrast, larger models such as Wan2.1 and Wan2.2 deliver strong absolute gains in SR, but their much longer inference times reduce their cost-performance, which may limit their practicality in real-time or resource-constrained deployments. In comparsion, lightweight, fast architectures like LTX-Video offer a promising direction for “Lightweight World Models” that can still provide meaningful planning benefits under strict compute budgets.
>
> ---
>
> ## References:
>
> [1] Amir Bar, Gaoyue Zhou, Danny Tran, Trevor Darrell, and Yann LeCun. Navigation world models. In Proceedings of the IEEE/CVF Conference on Computer Vision and Pattern Recognition (CVPR), 2025.
>
> [2] Shenyuan Gao, Siyuan Zhou, Yilun Du, Jun Zhang, and Chuang Gan. Adaworld: Learning adaptable world models with latent actions. In International Conference on Machine Learning (ICML), 2025.

---

> > ### Comment · Reviewer_8Zz4 · 2025-11-21
> > **Response to authors**
> >
> > The author's reply answered my question, and I kept my original positive score unchanged

---

> > > ### Author Response · Authors · 2025-11-24
> > > **Response to 8Zz4**
> > >
> > > Dear Reviewer 8Zz4,
> > >
> > > Thank you very much for your positive feedback and for acknowledging our efforts.
> > >
> > > If our recent response has clearly addressed your concerns and resolved the weaknesses you mentioned, we would be very grateful if you could consider updating your score accordingly.
> > >
> > > If not, we would also be happy to clarify any remaining questions or concerns further. Please let us know if there is anything else we can elaborate on.
> > >
> > > We sincerely appreciate your time and thoughtful evaluation. Thank you again.

---

### Official Review · Reviewer_76LM · 2025-10-31

**Soundness:** 3
**Presentation:** 3
**Contribution:** 3
**Rating:** 6
**Confidence:** 3

**Summary:**

The paper introduces WoW!, a new open platform to address the flaws of existing world model (WM) benchmarks, arguing that "closed-loop" embodied task success, not "open-loop" visual quality, is the proper metric to evaluate the WMs. The platform provides a unified planning strategy and action API to evaluate heterogeneous WMs on four categories of embodied tasks. Using this benchmark, the paper finds that visual quality does not correlate with task success, but controllability does, and that performance scales with both in-domain post-training data and inference-time compute.

**Strengths:**

1. The paper's main contribution is shifting the evaluation of world models from open-loop visual fidelity to closed-loop embodied task success. This is a significant and necessary service for the field.
2. The benchmark, through its Unified Action API, allows for direct, fair comparison of heterogeneous SOTA video generators across comprehensive embodied tasks.
3. The paper's "three surprises" are all well-supported by evidence and can provide insights for the community.

**Weaknesses:**

1. The "three surprises" are to some extent overstated, especially the first and the second ones, as these findings have been pointed
out by many previous papers, e.g. [1,2,3].
2. The paper does not directly specify the reward function used to score the world model's imagined trajectories.
3. The method on how to refine the policy during planning is not that clear (beyond directly using the plan with the highest score).
4. There already exist some benchmarks that also evaluate the pretrained video generation models' capability on control tasks, e.g. VP2 [3]. Some of the insights in WoW! are also pointed out in VP2, like the first "surprise". Although WoW! is indeed a more comprehensive benchmark for evaluating video generation models as WMs on embodied tasks, the paper must reference these prior works and include a discussion or comparison to properly situate WoW!'s contributions within the existing literature.

[1] Gao S, Zhou S, Du Y, et al. Adaworld: Learning adaptable world models with latent actions[J]. arXiv preprint arXiv:2503.18938, 2025. \
[2] He H, Zhang Y, Lin L, et al. Pre-trained video generative models as world simulators[J]. arXiv preprint arXiv:2502.07825, 2025. \
[3] Tian S, Finn C, Wu J. A control-centric benchmark for video prediction[J]. arXiv preprint arXiv:2304.13723, 2023.

**Questions:**

1. See 2 and 3 in the Weakness.
2. Regarding the limited gains in manipulation tasks, where the WM provides only a marginal boost over the VLM-only baseline as is pointed out, could you provide a deeper analysis? Is it purely a failure to model contact-rich physics, and what are potential solutions? Given that manipulation is a key focus of embodied AI, what other roles might WMs play in these complex tasks where their current planning utility appears minimal?

---

> ### Author Response · Authors · 2025-11-19
> **Response to Reviewer 76LM [1/4]**
>
> We appreciate the reviewer’s constructive feedback and would like to provide additional clarification.
>
> > **W1, W4**: Three surprises, especially the first and the second ones, have been pointed out by many previous papers e.g., VP2 [3]. And WoW! must reference these prior works and include a discussion or comparison to properly situate WoW!'s contributions within the existing literature.
>
> **TL;DR: While prior works have observed some of these phenomena, our benchmark provides a more comprehensive and systematic evaluation across multiple embodied tasks and diverse recent WMs. We have clarified these distinctions and added relevant citations in the revised manuscript.**
>
> We sincerely thank the reviewer for pointing out these relevant works. We agree with the reviewer that, although prior works have observed some of these phenomena, our benchmark provides a more comprehensive and systematic evaluation across multiple embodied tasks and diverse WMs. This is the main way in which our work goes beyond earlier studies that focus on specific tasks or model families. We highlight the differences as follows:
>
> - **Relation to concurrent works on action-controllable video generation.** Concurrent works such as Adaworld [1] and He et al. [2] mainly focus on extending the action control ability of video generation models via finetuning or new training recipes. Our findings, especially those on how to extend the action control ability of pretrained video generation models through post-training, are complementary to these works. In particular, we study how to finetune **multiple pretrained video generators** with **different dataset sizes** and under **minimal resource budgets**, and we analyze which configurations yield the most cost-effective performance gains on downstream embodied tasks for different base models (e.g., larger models such as Wan2.1/2.2 versus smaller models such as SVD and LTX-Video). This provides practical guidance for choosing and adapting video generation models as WMs under realistic resource constraints and dataset sizes.
>
> - **Scope and difficulty of tasks compared to VP2.** VP2 [3] is an important video prediction benchmark but primarily targets robotic manipulation with relatively simple tasks (e.g., pushing specific objects, opening drawers) and around 310 instances. In contrast, our platform evaluates a **broader** and **more complex** set of embodied tasks, including Active Recognition, Active EQA, Image-Goal Navigation, and Robotic Manipulation, with a total of 1,079 instances. These tasks require active perception, language grounding, long-horizon navigation, and more complex manipulation, which go beyond the settings considered in VP2.
>
> - **Breadth and nature of evaluated WMs.** Our benchmark evaluates a **more diverse set** of video generation models used as WMs, including 11 recent large-scale pretrained WMs (spanning 2021.06–2025.08), most of which are trained on large-scale Internet video datasets with different architectures and training recipes. VP2 [3], in contrast, mainly evaluates task-specific video prediction models (roughly 2019–2022) trained from scratch on a single robotic manipulation dataset. After the release of Sora in 2024.02 by OpenAI, there has been a major shift in training scale and model design for video generation, which motivates the need for a new, more comprehensive platform to evaluate these large-scale pretrained video generators as WMs for embodied AI.
>
> - **Evaluation suite and planning / reward design.** To support complex embodied tasks and heterogeneous WMs, we develop more advanced and modular evaluation suites. VP2 [3] uses a visual foresight-style policy that randomly samples proposal action plans and scores them using simple pixel-wise MSE between the predicted and goal states as the reward. Such a policy and reward design are less suitable for tasks where goal states are not directly observable or provided (e.g., Active EQA, Active Recognition, or more complex manipulation). In WoW!, we primarily follow recent advances in VLM-based reward models [4,5,6] to design more flexible and powerful proposal and revision policies, and we also report results for classic baselines (e.g., heuristic visual foresight-style policies and LPIPS-based reward functions similar in spirit to pixel-wise MSE in VP2). This allows us to incorporate recent progress in reward modeling and to provide more detailed insights into how to design effective model-based methods for complex embodied tasks using video generators as WMs.
>
> We have incorporated these clarifications, comparisons, and citations in the revised manuscript (Appendix A. Related Work, and 3.2 Ablation and Findings) to position our platform more clearly with respect to VP2 and other related works.

---

> ### Author Response · Authors · 2025-11-19
> **Response to Reviewer 76LM [2/4]**
>
> > **W2, Q1**: The paper does not directly specify the reward function used to score the world model's imagined trajectories.
>
> **TL;DR: In our framework, the reward function is implemented as one part of the revision policy. By default, it is a VLM-based reward model; in some tasks (e.g., Image-Goal Navigation) we also report results with an LPIPS-based reward.**
>
> We appreciate the reviewer's valuable comment and would like to provide clarification.
> First, we would like to do a short review.
> In our framework, the component that plays the role of a reward function is integrated into what we call the **revision policy**. This component takes as input the imagined trajectories (predicted future observations together with their corresponding action sequences) and outputs the final selected/revised action sequences or text answer, depending on the task. Throughout the paper, we use the term “revision policy” to emphasize that this component does more than just scoring trajectories: it refines the initial proposals from the proposal policy into a final decision. We apologize for the confusion caused by this difference in terminology from the standard model-based planning setup, and we will clarify this in the revised manuscript.
>
> In our implementation, motivated by recent advances in using VLMs as reward models [4, 5, 6], we mainly instantiate the revision policy as a VLM-based reward model. Concretely, we use Qwen2.5-VL-72B as the default reward model to evaluate imagined trajectories, as detailed in Appendix B.5 (Policies in Embodied Tasks). The VLM receives the imagined observations (and task context) and produces preferences that we use to rank and select the final action plan.
>
> In addition to VLM-based reward models, we also report results with task-specific reward functions when the goal state is provided in certain tasks. For example, in Image-Goal Navigation, where the goal image is accessible, we use an LPIPS-based reward that measures perceptual distance between predicted observations and the goal image. These results have been included in the section 3.2 Ablation and Findings of the revised manuscript and complement our main VLM-based results.
>
>
> > **W3, Q1**: The method on how to refine the policy during planning is not that clear (beyond directly using the plan with the highest score).
>
> **TL;DR: For Image-Goal Navigation and Robotic Manipulation, the revision policy follows a simple score-and-select scheme over imagined trajectories. For Active Recognition and Active EQA, the revision policy is richer: it both selects the next action and uses the predicted future observations to produce the final answer.**
>
> We appreciate the opportunity to clarify this. Due to space limits, we did not fully expand on the revision policy design in the main manuscript. Here we provide a more detailed description of how the revision policy operates across different embodied tasks.
>
> For Active Recognition (AR) and Active Embodied Question Answering (A-EQA), each episode combines **action planning** and **question answering**. In these tasks, world model predictions are used not only to guide action planning, but also to provide additional visual evidence for producing the final answer. For example, generated multi-view observations help reduce occlusion and improve recognition in AR. This differs from Image-Goal Navigation and Robotic Manipulation, where the objective is only to plan actions that reach a goal state, without producing a separate textual or categorical answer.
>
> As a result, in AR and A-EQA, the revision policy does more than selecting the plan with the highest score. It also takes the predicted future observations as input to an answer head. This is a different instantiation of $\pi_{\text{revision}}$ than the simple “score-and-select” form in Eq.~(4) of the main manuscript.
>
> For AR and A-EQA, we explicitly decompose the output of $\pi_{\text{revision}}$ at time step $t$ into an **action component** and an **answer component**. Let $\hat{y}_{t}$ denote the predicted answer at time $t$ (a category label for AR and a natural-language answer for A-EQA). We write:
> $$
> \\mathbf{D}^{\\star}\_t =
> \\bigl(\\hat{\\mathbf{A}}^{\\star}\_t, \\hat{y}\_t \\bigr) =
> \\pi\_{\\text{revision}}\\Bigl(
> \\lbrace (\\hat{\\mathbf{A}}\_t^{(m)}, \\hat{\\mathbf{O}}\_t^{(m)}) \\rbrace\_{m=1}^{M},
> \\mathbf{o}\_t, \\mathrm{g}
> \\Bigr).
> $$

---

> ### Author Response · Authors · 2025-11-19
> **Response to Reviewer 76LM [3/4]**
>
> In our implementation, the action component $\hat{\mathbf{A}}^{\star} _{t}$ is still chosen by a score-and-select rule using an action scoring function $S _{\text{act}}$, while the answer component $\hat{y} _{t}$ is obtained by aggregating predicted futures from all candidates:
>
> $$
> \\hat{\\mathbf{A}}^{\\star}\_t =
> \\hat{\\mathbf{A}}\_t^{(m^{\\star})},
> \\quad\\text{where}\\quad
> m^{\\star} =
> \\text{arg\\,max}\_{m \\in \\lbrace 1,\\dots,M \\rbrace}
> S\_{\\text{act}}\\Bigl(
> \\hat{\\mathbf{A}}\_t^{(m)},\\,\\hat{\\mathbf{O}}\_t^{(m)}
> \\,\\big|\\, \\mathbf{o}\_t,\\,\\mathrm{g}
> \\Bigr),
> $$
> $$
> \\hat{y}\_t =
> f\_{\\text{ans}}\\Bigl(
> \\mathbf{o}\_t,\\,\\mathrm{g},
> \\,\\lbrace\\hat{\\mathbf{O}}\_t^{(m)}\\rbrace\_{m=1}^{M}
> \\Bigr).
> $$
>
> Here, $S _{\text{act}}(\cdot)$ is an action scoring function (e.g., favoring trajectories that move the agent toward more informative views or closer to the target object), and $f _{\text{ans}}(\cdot)$ is an answer head that uses the current observation $\mathbf{o} _{t}$, the goal $\mathrm{g}$, and the set of predicted futures $\{\hat{\mathbf{O}} _{t}^{(m)}\} _{m=1}^{M}$ as multi-view evidence. In practice, $f _{\text{ans}}$ is implemented with a vision-language model that takes the frames as input.
>
> Thus, for AR and A-EQA, the revision policy operates in two coupled ways: it (i) selects how the agent should move next via $S _{\text{act}}$ and $\hat{\mathbf{A}}^{\star} _{t}$, and (ii) uses the simulated rollouts as additional context to produce a more informed answer $\hat{y} _{t}$. This is a richer instantiation of $\pi _{\text{revision}}$ than the pure score-and-select form in Eq.~(4), which only chooses the highest-scoring action plan. We have clarified this distinction, along with the AR and A-EQA-specific form of $\pi _{\text{revision}}$, in the Appendix (B.5 Policies in Embodied Tasks) of the revised manuscript.
>
>
> > **Q2**: In manipulation tasks, could you provide a deeper analysis? Is it purely a failure to model contact-rich physics, and what are potential solutions?
>
> **TL;DR: We summarize two major factors for the failure: limitations of WMs for contact-rich physics, and limitations of revision policies. Potential solutions include physics-guided motion generation and post-training video generators with video reward models.**
>
>
> *1. Deeper Analysis for Manipulation Tasks*
>
> Thanks for the insightful question.
> We agree that the failure cases in manipulation deserve deeper analysis. We summarize two major factors below.
> - **Limitations of WMs for contact-rich physics.** Our current evidence suggests that a major factor is the limited ability of existing video generation models to accurately capture contact-rich physical interactions, which are essential for successful manipulation. Most current video generators are trained with objectives that primarily emphasize pixel-level similarity between generated and ground-truth videos, without explicitly modeling underlying physical dynamics, as also discussed in concurrent work [7,8]. As a result, the WMs can produce rollouts that look visually plausible but are physically inconsistent (e.g., unrealistic object motion or incorrect contact responses), which in turn can mislead planning and control.
> - **Limitations of revision policies.** On the decision-making side, VLM-based revision policies may also lack sufficient sensitivity to such subtle physical violations in imagined trajectories. They can assign high scores to sequences that are visually reasonable but physically incorrect, which contributes to sub-optimal action selection in contact-rich regimes.
>
> *2. Potential Solutions*
>
> As we discuss in the manuscript, there are several promising directions to address these issues on the world model side. One line of work is **physics-guided motion generation**, where video generation is regularized via simulator-in-the-loop schemes [11,12]. These methods aim to better capture contact, compliance, friction, and state changes of articulated or deformable objects. However, the heuristic or manually designed aspects of the underlying physics simulators may limit scalability and generalization to complex real-world settings.
>
> Another direction is to **post-train video generators with video reward models** in a reinforcement learning style paradigm [9,10]. In this case, a learned reward model, trained to assess physical realism and task relevance, provides feedback that shapes the video generator toward trajectories that better respect physical laws. This can help bridge the gap between appearance-based video generation and physically grounded world modeling, and we see it as a natural next step for improving WMs in contact-rich manipulation tasks.

---

> ### Author Response · Authors · 2025-11-19
> **Response to Reviewer 76LM [4/4]**
>
> ## References:
>
> [1] Gao S, Zhou S, Du Y, et al. Adaworld: Learning adaptable world models with latent actions[J]. arXiv preprint arXiv:2503.18938, 2025.
>
> [2] He H, Zhang Y, Lin L, et al. Pre-trained video generative models as world simulators[J]. arXiv preprint arXiv:2502.07825, 2025.
>
> [3] Tian S, Finn C, Wu J. A control-centric benchmark for video prediction[J]. arXiv preprint arXiv:2304.13723, 2023.
>
> [4] Gershom Seneviratne, Jianyu An, Sahire Ellahy, Kasun Weerakoon, Mohamed Bashir Elnoor, Jonathan Deepak Kannan, Amogha Thalihalla Sunil, and Dinesh Manocha. Halo: Human preference aligned offline reward learning for robot navigation. ArXiv, 2508.01539, 2025.
>
> [5] Yibin Wang, Yuhang Zang, Hao Li, Cheng Jin, and Jiaqi Wang. Unified reward model for multimodal understanding and generation. ArXiv, 2503.05236, 2025a.
>
> [6] Jie Wu, Yu Gao, Zilyu Ye, Ming Li, Liang Li, Hanzhong Guo, Jie Liu, Zeyue Xue, Xiaoxia Hou, Wei Liu, Yan Zeng, and Weilin Huang. Rewarddance: Reward scaling in visual generation. ArXiv, 2509.08826, 2025.
>
> [7] Chenyu Li, Oscar Michel, Xichen Pan, Sainan Liu, Mike Roberts, and Saining Xie. PISA experiments: Exploring physics post-training for video diffusion models by watching stuff drop. ArXiv, 2503.09595, 2025a.
>
> [8] Bingyi Kang, Yang Yue, Rui Lu, Zhijie Lin, Yang Zhao, Kaixin Wang, Gao Huang, and Jiashi Feng. How far is video generation from world model: A physical law perspective. ArXiv, 2411.02385, 2024.
>
> [9] Jie Wu, Yu Gao, Zilyu Ye, Ming Li, Liang Li, Hanzhong Guo, Jie Liu, Zeyue Xue, Xiaoxia Hou, Wei Liu, Yan Zeng, and Weilin Huang. Rewarddance: Reward scaling in visual generation. ArXiv, 2509.08826, 2025.
>
> [10] Jie Liu, Gongye Liu, Jiajun Liang, Ziyang Yuan, Xiaokun Liu, Mingwu Zheng, Xiele Wu, Qiulin Wang, Wenyu Qin, Menghan Xia, Xintao Wang, Xiaohong Liu, Fei Yang, Pengfei Wan, Di Zhang, Kun Gai, Yujiu Yang, and Wanli Ouyang. Improving video generation with human feedback. ArXiv, 2501.13918, 2025.
>
> [11] Ke Zhang, Cihan Xiao, Yiqun Mei, Jiacong Xu, and Vishal M. Patel. Think before you diffuse: Llms-guided physics-aware video generation. ArXiv, 2505.21653, 2025b
>
> [12] Chen Wang, Chuhao Chen, Yiming Huang, Zhiyang Dou, Yuan Liu, Jiatao Gu, and Lingjie Liu. Physctrl: Generative physics for controllable and physics-grounded video generation. ArXiv, abs/2509.20358, 2025a

---

### Author Response · Authors · 2025-11-26
**Reviewer Comments Summary**

We sincerely thank all the reviewers for their careful and constructive feedback.
In this rebuttal, we have made every effort to address all concerns and have substantially revised and expanded the manuscript.
Our updated manuscript now includes **6 new sections and subsections** featuring theoretical justifications, extended discussions, and **4 additional experiments**. Major updates include:

* **Clearer positioning w.r.t. prior work** (e.g., VP2 and concurrent world-model / video-generation benchmarks), including an explicit comparison of tasks, model coverage, and evaluation scope.
* **Clarified planning framework and reward / revision policy**, making explicit that the reward is implemented via a VLM-based revision policy, and detailing task-specific variants, including LPIPS-based rewards for ImageNav and joint action+answer revision for AR and A-EQA.
* **Isolating the contribution of world models**, by adding experiments with simple heuristic proposal policies and simpler revision policies (e.g., pure LPIPS), showing that WMs still provide strong gains beyond powerful VLM-only baselines.
* **New cross-domain experiments**, where WMs post-trained on synthetic HSSD data are evaluated on HM3D/Matterport3D AR and ImageNav, demonstrating that WMs still improve performance under domain shift while highlighting remaining generalization gaps.
* **Deeper analysis of manipulation tasks and failure modes**, emphasizing limitations of current video generators for contact-rich physics, challenges for VLM-based reward models, and discussing future directions such as physics-guided generation and video reward models.
* **Clarifications on the unified action API and control alignment**, including details on how we map structured actions where possible, how we design and validate prompts for text-controlled WMs, and why action misalignment is limited in practice.
* **Cost–performance analysis of WMs**, measuring inference-time cost, reporting success-rate gain per unit time, and showing that lightweight models achieve the best cost–performance while large models provide higher absolute SR but with higher latency.

All major changes are highlighted in the revised manuscript. We hope these revisions address the reviewers’ main concerns and help foster further constructive discussion during the decision-making phase. We sincerely appreciate the reviewers' understanding and patience.

---

### Meta-Review · Area_Chair_krjm · 2026-01-04

**Summary:**

Reviewers were generally positive about this work, with concerns focusing primarily on clarity and positioning rather than the core contribution. Some reviewers noted that parts of the main findings had precedents in prior work and raised questions about the clarity of the planning and reward formulation, attribution of gains to world models versus strong VLM-based policies, manipulation performance, generalization, and computational cost. These issues were largely addressed in the rebuttal through clearer exposition, stronger positioning with respect to prior work, additional ablations, cross-domain experiments, deeper analysis, and cost–performance evaluations. I recommend acceptance because the work is timely and introduces a much needed diagnostic framework that directly addresses an important question in the field, namely whether and how video based world models lead to higher embodied task success. Overall, this paper represents a good step forward for embodied AI and world model research.

**Reviewer Concerns:**

The rebuttal effectively addressed the major reviewer concerns around clarity and attribution. In particular, concerns about unclear planning and reward formulations were resolved through clearer explanations of the revision policy. Doubts about whether gains were driven by strong VLM policies rather than world models were addressed with new ablations using weaker proposal policies and simpler reward functions. Questions about generalization were partially addressed via added cross domain transfer experiments. Concerns about manipulation performance were addressed with deeper failure analysis and discussion. Computational cost concerns were mitigated by a new cost performance analysis highlighting lightweight models. Remaining concerns are relatively minor and mostly editorial or scope related, such as the fact that generalization is still limited to a small number of domains and simulators, and that detailed inference cost breakdowns could be more prominently integrated into the main text rather than appendices. Overall, no outstanding issues remain that undermine the core claims or contributions of the work.

**Reviewer Scores:**

76LM -> Maintained the positive score
8Zz4 -> Maintained the positive score
KANk -> Maintained the positive score
Bvsj -> Maintained the positive score

---

### Decision · Program_Chairs · 2026-01-26

Accept (Oral)